# MoA: Mixture of Sparse Attention for Automatic Large Language Model Compression

## Abstract

Sparse attention can effectively mitigate the significant memory and throughput demands of Large Language Models (LLMs) in long contexts. Existing methods typically employ a uniform sparse attention mask, applying the same sparse pattern across different attention heads and input lengths. However, this uniform approach fails to capture the diverse attention patterns inherent in LLMs, ignoring their distinct accuracy-latency trade-offs. To address this challenge, we propose the Mixture of Attention (MoA), which automatically tailors distinct sparse attention configurations to different heads and layers. MoA constructs and navigates a search space of various attention patterns and their scaling rules relative to input sequence lengths. It profiles the model, evaluates potential configurations, and pinpoints the optimal sparse attention compression plan. MoA adapts to varying input sizes, revealing that some attention heads expand their focus to accommodate longer sequences, while other heads consistently concentrate on fixed-length local contexts. Experiments show that MoA increases the effective context length by $3.9\times$ with the same average attention span, boosting retrieval accuracy by $1.5 - 7.1\times$ over the uniform-attention baseline across Vicuna-{7B,13B}, and Llama3-{8B,70B} models. Moreover, MoA narrows the capability gaps between sparse and dense models, reducing the maximum relative performance drop from $9\% - 36\%$ to within $5\%$ across two long-context understanding benchmarks. MoA achieves a $1.2 - 1.4\times$ GPU memory reduction, boosting decode throughput by $6.6 - 8.2\times$ and $1.7 - 1.9\times$ over FlashAttention2 and vLLM, with minimal performance impact.

## 1 Introduction

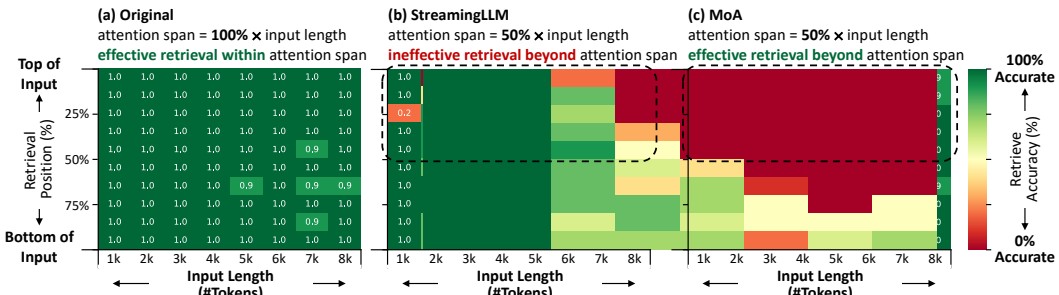

Figure 1: Retrieval accuracy of the Vicuna-7B model using different attention methods across varying input lengths and retrieval positions on the LongEval benchmark (Li et al., 2023a). This retrieval benchmark takes massive key-value pairs as inputs and tests the accuracy to retrieve values based on given keys from diverse positions. (a) Original model with a full attention span; (b) StreamingLLM with half the attention span, showing reduced effectiveness beyond the span; (c) MoA with half the attention span, maintaining effectiveness beyond the span.

Large Language Models (LLMs) exhibit remarkable versatility across numerous applications (Brown et al., 2020; Tay et al., 2022; Wan et al., 2023). Central to LLM is the attention mechanism (Vaswani et al., 2017), which computes interactions among tokens within a certain span, thereby enabling context understanding. Scaling input length is crucial for enhancing LLM capabilities (Chen et al., 2023; Tworkowski et al., 2023), including fact retrieval, summarization, few-shot learning, question

answering and so on (Bai et al., 2023; Yuan et al., 2024). However, the ever-growing attention computation and Key-Value Cache (KV-Cache) pose significant efficiency challenges (Sheng et al., 2023; Xiao et al., 2024c; Han et al., 2023; Kwon et al., 2023).

Previous work proposes sparse attention methods to address the efficiency challenges of long contexts in generative LLMs. These methods typically employ a uniform, fixed-span sliding window mask across all heads and input lengths, limiting attention to local contexts only (Xiao et al., 2024c; Han et al., 2023). This approach allows the LLM to take long inputs with a fixed attention span, keeping bounded attention computation and KV caching overhead. Following previous works (Chen et al., 2023; Tworkowski et al., 2023), we quantify the effective context length as the maximum input length where content retrieval accuracy exceeds a 90% threshold. In principle, fixed-span local attention can gradually aggregate global information through multiple model layers, yielding a longer effective context length than each attention span (Feng et al., 2022; Zaheer et al., 2020). Nonetheless, we reveal that uniform masks, like StreamingLLM (Xiao et al., 2024c), hardly extend effective context length beyond the span, as shown in Figure 6. Figure 1(b) further illustrates such limitation: with a 50% attention span mask, StreamingLLM fails to accurately retrieve content from the earlier half of the input and performs even worse at longer input lengths. Figure 2 reveals one possible explanation for the problem: while some attention heads focus on local contexts, others encompass the broad span of the entire input sequence. Consequently, the uniform approach fails to achieve a long effective context length as it limits the attention span of the global-context heads, while excessively allocates compute and memory budget for local-context heads. Additionally, as the input length increases, some attention heads need a faster increase in attention span than others to avoid serious performance degradation, as shown in Table 1. Unfortunately, the uniform approaches do not include heterogeneous rules to scale the attention spans differently for various heads. Besides, existing model compression methods (Men et al., 2024; Lin et al., 2023; Xiao et al., 2024b; Li et al., 2024a; Kim et al., 2023; Li et al., 2024b) use *general language modeling corpora* to decide the compression plan, which cannot accurately profile the influence of compression on *long-context tasks*.

In this work, we propose Mixture of Attention (MoA), a training-free sparse attention method. As illustrated in Figure 3, MoA constructs the search space of heterogeneous elastic rules of attention spans. For automatic LLM compression, MoA first utilizes gradient-based profiling to inspect the influences of each attention position on the prediction loss. Based on the profiling results, MoA tailors heterogeneous sparse attention configurations for each model layer and attention head. During profiling, MoA employs a calibration dataset with long-range dependencies and uses the original dense model's response instead of the human-written response as the reference to calculate the loss. This ensures an accurate profiling of the attention influences to facilitate better compression results. Our contributions are summarized as follows.

- **Heterogeneous Elastic Rules**. We propose heterogeneous elastic rules for masks of each attention head. We formulate MoA compression search space to include a diverse range of elastic rules that tailor the local attention span relative to the input length for each attention head. The heterogeneous elastic rules improve the fact retrieval accuracy of MoA from 25% to 98% compared with masks with uniform span and scaling function for each head.

- **Calibration Dataset Construction** We emphasize the importance of data engineering in LLM compression. Our findings demonstrate that, instead of relying on general language modeling datasets and human responses, using datasets with long-range dependencies and referencing the original LLM's responses is essential for accurately profiling the effects of compression.

- **Automatic Optimization**. We propose an automatic pipeline to find the optimal compression plan encompassing heterogeneous elastic rules for various attention heads. This pipeline can efficiently find the optimal plan within several hours, for example, two hours for compressing Vicuna-13B.

Experiments show that MoA achieves $6.6 - 8.2\times$ throughput improvements over FlashAttention2, $1.7 - 1.9\times$ over vLLM framework on 7B and 13B dense LLMs at a 50% density (the average of KV-Cache length / input length), with only a 1% average relative degradation in retrieval accuracy. The significant throughput improvements of MoA over FlashAttention2 can be attributed to four factors: (1) the static size of the KV-cache, (2) reduced attention computations, (3) increased batch size

enabled by reduced memory usage, and (4) a specialized kernel implementation. Additionally, MoA achieves over 90% retrieval accuracy with just 25% average density, far surpassing sparse attention baselines that need a density of 75% to 100% for similar performance. On long-context understanding benchmarks, MoA performs comparably to dense models, with a maximum relative performance drop of less than 5%, which is about one-sixth of that observed with the uniform sparse attention baseline. Our code is available at `https://anonymous.4open.science/r/MoA-Review`.

## 2 PRELIMINARY AND RELATED WORK

### 2.1 ATTENTION MECHANISM

The Multi-Head Self Attention (MHA) mechanism (Vaswani et al., 2017) is crucial to the functionality of LLMs. It starts with an input sequence transformed into query (Q), key (K), and value (V) matrices through linear projections. These matrices, combined with the cached K and V (KV-Cache) from previous sequences, compute the attention matrix (A). This calculation is modified by a causal mask (M) to ensure autoregressive properties, resulting in the output (O), as depicted in Equation 1:

$$\mathbf{S} = \mathbf{Q}\mathbf{K}^T, \quad \mathbf{A} = \text{softmax}(\mathbf{S} + \mathbf{M}), \quad \mathbf{O} = \mathbf{A}\mathbf{V} \tag{1}$$

Autoregressive inference in LLMs involves two stages: prefill and decode. During prefill, the model processes the entire input sequence to generate the initial response token. In the subsequent decode stage, it uses the newly generated token and previously cached K and V matrices to produce subsequent tokens until the generation concludes. Although effective, this iterative process increases memory and computation demands due to the expanding KV-Cache.

### 2.2 EFFICIENT ATTENTION

Efficient methods are proposed to mitigate the computation and memory costs associated with attention. One branch of work uses dynamic sparse attention masks to adaptively skip attention computations during prefill stage (Pagliardini et al., 2023; Qu et al., 2022; Roy et al., 2021; Wang et al., 2021; Lu et al., 2021; Kitaev et al., 2020) or drop KV-Cache during decode stage (Anagnostidis et al., 2023; Zhang et al., 2023; Ge et al., 2023; Sheng et al., 2023; Liu et al., 2023a) based on the input sequences. However, due to the complex control and computation flow, dynamic prefill often requires specific hardware to achieve substantial wall-time speedup (Qu et al., 2022; Wang et al., 2021; Lu et al., 2021; Ham et al., 2021; 2020). Additionally, dynamic KV-Cache pruning in the decode stage may require extensive retraining (Anagnostidis et al., 2023), additional KV-Cache score computation (Sheng et al., 2023; Zhang et al., 2023; Liu et al., 2023a; Ge et al., 2023; Li et al., 2024c; Cai et al., 2024), or extensive memory swap for KV-Cache retrieval (Tang et al., 2024b; Xiao et al., 2024a).

Another branch of work uses static sparse attention, where predefined masks are applied consistently across all processed sentences. Thanks to the fixed computation flow, static sparse attention is generally more efficient and GPU-friendly. For language understanding models such as BERT (Devlin et al., 2018), various masks are used (Zaheer et al., 2020; Beltagy et al., 2020; Child et al., 2019; Zhou et al., 2024; Xiao et al., 2024c; Han et al., 2023). But for generative LLMs, the predominant method is the fixed-span sliding window mask with global attention on a few initial tokens (Xiao et al., 2024c; Han et al., 2023). With the local attention pattern, the KV-Cache beyond the current attention span can be dropped, saving much memory for long sequence scenarios. However, the uniform static masks across different attention heads and input lengths are model- and data-agnostic, which can compromise LLMs' effective context length and lead to suboptimal performance in long sequence scenarios. Our method falls within this category, benefiting from the efficiency and training-free advantages, while addressing the performance limitations encountered by previous methods.

In addition to sparse attention, alternative mechanisms have been proposed to replace traditional attention for long-sequence modeling (Gu & Dao, 2023; Peng et al., 2023; Sun et al., 2023; Poli et al., 2023; Li et al., 2022; Kacham et al., 2023; Peng et al., 2021; Choromanski et al., 2020; Wang et al., 2020). However, these new mechanisms often require different weights compared to vanilla transformers, imposing significant re-training overhead for LLMs.

Previous works also propose LLM acceleration frameworks (Gugger et al., 2022; Aminabadi et al., 2022; Sheng et al., 2023; Kwon et al., 2023), as well as kernel-level optimizations (Dao et al., 2022;

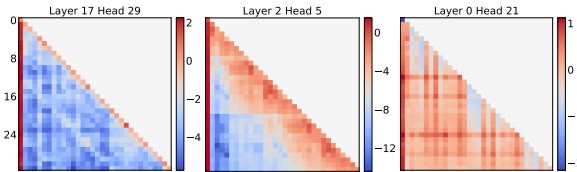

Figure 2: Examples of attention matrices from different attention heads of the Vicuna-7B model. Each attention matrix is averaged over 256 data items from the LongEval dataset.

| Layers | Window/Input Len. | | |
|---|---|---|---|
| | 2k/4k | 2k/8k | 4k/8k |
| 6, 7, 8 | 0.83 | 0.29 | 0.61 |
| 9, 10,11 | **0.99** | 0.81 | 0.96 |
| 17,18,19 | 0.97 | **0.94** | **0.97** |

Table 1: Retrieval accuracy of Vicuna-7B with sliding-window sparse attention across various model layers, window spans, and input lengths.

Dao, 2023; Shah et al., 2024). These kernel and system optimizations are orthogonal to our work and can be integrated to further enhance efficiency.

# 3 MIXTURE OF ATTENTION (MOA)

We first illustrate the heterogeneity of the attention patterns in pre-trained LLMs in Section 3.1. Based on this insight, we define the search space for our Mixture-of-Attention (MoA) method in Section 3.2.

## 3.1 MIXTURE OF ATTENTION PATTERNS AND ELASTIC RULES

**Heterogeneous Attention Patterns**. Different attention heads in LLMs exhibit heterogeneous attention patterns, as shown in Figure 2. For example, the first head primarily focuses on local contexts with a narrow-span sliding window, while the third head covers nearly the entire input, indicating global attention. The attention spans of different heads mostly remain constant across various tasks and datasets, as shown in Appendix D.1. Table 1 demonstrates that applying the same sliding-window sparse attention mask across model layers can lead to a 65% variance in retrieval accuracies. It conforms to the multi-head self-attention design principle of capturing varied information (Vaswani et al., 2017), as well as the findings from concurrent research that identifies specific attention heads for global text retrieval (Wu et al., 2024).

**Heterogeneous Elastic Rules**. In addition to heterogeneity at a certain length, different attention heads also exhibit varying elastic behaviors as the input length changes. Figure 2 illustrates this variability: for shorter inputs (the upper left part of the attention matrix), the second and third heads initially show global attention. However, as input length increases, the second head remains the medium-span local focus, while the third head continues to expand as global attention. Table 1 further evidences the diverse elastic rules. For example, at 4k input length, a 2k sliding-window sparse attention mask on layers 9 to 11 yields better retrieval accuracy than on layers 17 to 19. However, the opposite is true for an 8k input length. This data supports the visual observations from Figure 2, highlighting that attention patterns respond to input length scaling differently. Leveraging these insights, MoA encompasses heterogeneous elastic rules as the search space.

## 3.2 HETEROGENEOUS ELASTIC RULE SEARCH SPACE

In designing the search space for the MoA mask, we consider the inherently heterogeneous and elastic nature of LLM attention patterns. As shown in Figure 3(a), we adopt a hardware-friendly sliding-window mask as our base sparse attention mask (Beltagy et al., 2020). Following previous work (Xiao et al., 2024c; Han et al., 2023), the initial few tokens (64 tokens for MoA) are not masked. The attention span equals the sliding-window-span plus the number of initial unmasked tokens. We define the attention span $S$ of head $h$ at input length $N$ using a straightforward linear function:

$$S_h = \alpha_h + \beta_h \cdot N, \tag{2}$$

where $\alpha_h$ and $\beta_h$ are hyperparameters that control the base span and its expansion rate with input length of a specific attention head.

The $\alpha$ and $\beta$ hyperparameters for each attention head are chosen from multiple discrete options. By default, MoA uses 6 and 9 options for $\alpha$ and $\beta$, respectively. For LLMs with many heads and layers, the search space can become quite large. For example, for a 7B model consisting of 32 attention heads and 32 layers, the potential search space expands to $54^{1024}$ configurations. Thus, we design the automatic pipeline to efficiently pinpoint the optimal $\alpha$s and $\beta$s for any LLM.

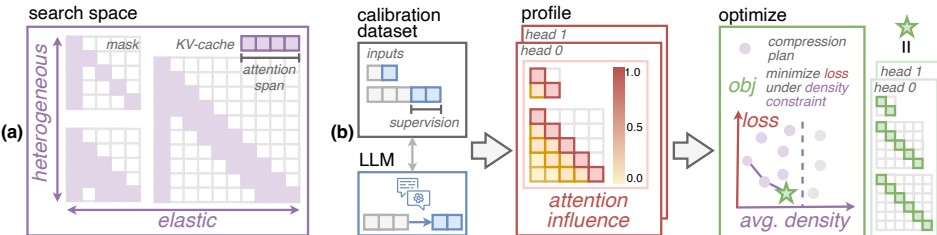

Figure 3: Overview of the MoA. (a) The sparse attention search space includes heterogeneous elastic rules of the attention span on sliding-window masks. (b) The automatic compression pipeline begins with a calibration dataset, which includes long-dependency contexts and supervision texts generated by the original dense LLM. MoA profiles each attention value's impact on model predictions within this dataset, revealing accuracy losses for different candidate elastic rules across various input lengths. The final optimization step selects elastic rules for each attention head to minimize the total prediction loss while adhering to specified density constraints.

## 4 Automatic Pipeline for MoA Compression

This section outlines the MoA automatic compression pipeline as shown in Figure 3(b). Starting with a trained LLM and a calibration dataset, MoA first **profiles** the influence of each attention value on the model's prediction loss for various input sequences from the calibration dataset. The masked sum of the influences represents the accuracy loss associated with each mask at different input lengths, showing the accuracy loss each candidate elastic rule could cause at that length. Then, MoA **optimizes** the compression plan by selecting the optimal elastic rule for each head, which minimizes the accuracy loss across various lengths while adhering to specified density constraints. The following sections provide detailed discussions of each step in this pipeline.

### 4.1 Attention Influence Profiling

In the profile step, MoA quantifies the impact of individual attention values on the final prediction loss of a pre-trained LLM. It informs the subsequent step about the influence of masking each attention value, revealing the accuracy trade-offs of the candidate elastic rules for each attention head.

The influence of each attention value is derived from the attention matrix $\mathbf{A}$ and its gradient $\partial L/\partial \mathbf{A}$, computed over a calibration dataset. When applying sparse attention masks, we approximate the change in the model's prediction loss, $\Delta L$, using a first-order Taylor expansion based on variations in the attention matrices $\mathbf{A}$: $\Delta L = \sum_h \sum_i \sum_j \partial L/\partial A_{h,i,j} \cdot \Delta A_{h,i,j}$. Here, $h$ indexes the attention heads across all layers, and $i, j$ are the row and column indices within each attention matrix $\mathbf{A}_h$. Details on the calibration dataset and the prediction loss $L$ are provided in Section 5.

We define the *attention influence* matrix, $E_{h,i,j}$, as the estimated change in loss, $\Delta L$, if the attention value $A_{h,i,j}$ is masked (i.e., set to zero). As shown in Equation 3, this measure considers both the direct and indirect effects of the mask. For notation simplicity, we omit the head index $h$ here. Initially, masking directly reduces the attention value to zero, represented by $\Delta A_{i,j|j} = -A_{i,j}$. Additionally, the softmax function in attention normalizes the sum of each row in the attention matrix to one. Thus, setting one attention value at column $j$ to zero causes an increase in the other attention values, $\Delta A_{i,n|j}, n \neq j$, within the same row. These two effects are integrated into the following formulation, whose derivation is provided in Appendix D.2:

$$E_{i,j} = \sum_n \frac{\partial L}{\partial A_{i,n}} \cdot \Delta A_{i,n|j} = \frac{\partial L}{\partial A_{i,j}} \cdot (-A_{i,j}) + \sum_{n \neq j} \frac{\partial L}{\partial A_{i,n}} \cdot A_{i,n} \cdot \frac{A_{i,j}}{1 - A_{i,j}} \quad (3)$$

In practice, we use backpropagation on a calibration dataset to calculate the average attention influence $\bar{\mathbf{E}}_h$ of each head across data items. The average attention influence is calculated respectively for different input lengths. The gradient $\partial L/\partial \mathbf{A}_h$ is computed using chain derivative in deep learning frameworks like PyTorch (Paszke et al., 2019). The detailed calibration dataset setup is discussed in Section 5.

With the average attention influence of each head, MoA can calculate the accuracy loss of applying a candidate elastic rule at a specific input length. The loss is calculated with the sum of masked

Table 2: Calibration dataset design choices: dataset content, supervision, and response reference. Calibration dataset with long dependency and model alignment improves MoA performance on retrieval accuracy and perplexity. All tests are done at 25% average density at 8k input length.

| Dataset | Supervision | Reference | Long Dep. | Align Model | Retrieval Acc. ↑ | PPL ↓ |
|---------|-------------|-----------|-----------|-------------|------------------|-------|
| RedPajama | Context | - | ✗ | ✗ | 0.25 | 4.95 |
| MultiNews | Context & Summary | Human | ✗/✓ | ✗ | 0.27 | 4.62 |
| MultiNews | Summary | Human | ✓ | ✗ | 0.87 | 3.97 |
| MultiNews | Summary | Model | ✓ | ✓ | **0.95** | **3.96** |

average attention influence according to the rule. We denote $\mathbf{M}_{r_h}$ as the binary mask at head $h$ that corresponds to rule $r$, with masked positions marked as 1 and others as 0. We formalize accuracy loss $\Delta L$ as follows:

$$\Delta L = \sum_h \Delta L_{h,r_h} = \sum_h \sum_i \sum_j M_{r_h,i,j} \cdot \bar{E}_{h,i,j}. \tag{4}$$

After the profile stage, MoA acquires the unique accuracy-density trade-offs of elastic rules. It informs the allocation of denser masks to more sensitive heads and lighter masks to less sensitive ones. Profiling at different input lengths enables the identification of the most effective elastic rules, even for unseen lengths.

## 4.2 Automatic Optimization

MoA automatically selects the optimal elastic rule for each attention head to minimize accuracy losses across various sequence lengths under density budgets. Based on the profiling results, MoA first identifies Pareto front compression plans where any improvement in accuracy loss at one profile length would worsen another. To ensure the best generalization to lengths beyond those profiled, MoA then selects the plan that yields the minimum loss at an unseen length among the Pareto front solutions as the final plan.

Specifically, we utilize multi-objective optimization to search for a set of Pareto optimal compression plans across the profiled lengths. The objective for each length is to minimize the total accuracy loss while conforming to any user-defined density constraints. The objective is formulated as follows:

$$\arg\min_{r_h \in \mathbb{R}} \Delta L^{(N_i)}, N_i \in \mathbb{N}_{\text{profile}} \quad \text{s. t.} \frac{1}{H} \sum_{h=1}^{H} d_{r_h}^{(N_i)} \leq d_{\text{constr}}^{(N_i)}, \forall N_i \in \mathbb{N}_{\text{constr}}. \tag{5}$$

Here, superscript $(N)$ denotes values at different lengths; $\mathbb{N}_{\text{profile}}$ and $\mathbb{N}_{\text{constr}}$ denote the sets of lengths for profiling and those subject to density constraints, respectively; $\mathbb{R}$ denotes the set of candidate rules; $\Delta L^{(N_i)}$ denotes the accuracy loss due to compression; $d_{r_h}^{(N_i)}$ denotes the density of rule $r_h$ at head $h$; $d_{\text{constr}}^{(N_i)}$ denotes the average density constraint; $H$ denotes the total number of attention heads.

Such formulation corresponds to the classic multi-objective mixed-integer-programming problem, which can be effectively solved within minutes using existing linear solvers, like Gurobi (Gurobi Optimization, LLC, 2023). The detailed formulation and solving strategies are discussed in Appendix D.3.

Among the Pareto optimal compression plans, we select the one with the minimum loss at the unseen validation length as the optimal solution. This approach allows us to avoid profiling at every possible length while increasing the likelihood that the plan will generalize effectively to unseen lengths.

Thanks to this automatic pipeline, we efficiently get the elastic rules tailored for each attention head. With the pipeline, MoA minimizes the accuracy loss caused by attention sparsification, while conforming to user-defined density constraints.

## 5 Dataset and Supervision

In this section, we highlight the overlooked importance of calibration dataset design and its supervision objective in LLM compression. Calibration datasets are essential for sensitivity analysis across various

compression techniques, including weight pruning (Men et al., 2024; Lee et al., 2024; Liu et al., 2023b) and quantization (Lin et al., 2023; Xiao et al., 2024b; Li et al., 2024a; Kim et al., 2023). In this work, MoA profiles the attention influence on the calibration dataset, which is crucial for subsequent automatic optimization.

**Current Approach**. General language modeling datasets, such as human-written text corpus RedPajama (Computer, 2023), are commonly used as the calibration dataset. These datasets, supervised by next-token-prediction on the entire corpus, primarily capture attention patterns coherent with immediately preceding tokens. However, they lack long context dependencies, failing to address the global attention crucial for tasks like long-range retrieval.

Moreover, a notable misalignment exists between the model response and the human-written supervision. Consequently, it leads to inaccuracies when using human responses to compute attention values and gradients during profiling. For example, given the same question, a human might answer 'Blue', while the model could generate 'The blue color'. Using the human answer for supervision, attention influence is inaccurately quantified based on probability shift for predicting 'Blue'; this diverges from the objective of maintaining crucial attention for the original model prediction, 'The'. These inconsistencies arise from various factors, including mismatched positions, tones, and synonyms.

**MoA's Approach**. MoA enhances the calibration dataset by integrating *long-range dependencies* and *model alignment*. Specifically, we utilize the long-contextual MultiNews dataset (Fabbri et al., 2019), which includes summaries heavily dependent on long-range content. The summaries are generated by the original dense model and serve as supervision. Compared to current approaches that adopt human responses as the reference to calculate the cross-entropy loss $L$, using the responses generated by the original model as the supervision can facilitate accurate profiling, thus benefiting the compression.

**Approach Comparison**. We evaluate our design's effectiveness by varying dataset choices, supervision types, and summary references, while standardizing data item count and length to 50 and 8k words, respectively. Additional setups and evaluations are in Appendices A and B.3.1.

We show the importance of long-range dependencies by comparing the MoA compression plan generated with different datasets and supervisory methods. In Table 2, RedPajama (Computer, 2023) represents the general language modeling dataset, while MultiNews (Fabbri et al., 2019) highlights long-range contexts by aggregating multiple documents on a single incident. Additionally, each MultiNews item includes a human-written summary, providing even stronger long-range dependencies and better performance. Calculating loss on the summary of MultiNews leads to significantly better performance, with a 60% increase in retrieval accuracy and a 0.98 decrease in perplexity.

Furthermore, using summaries generated by the original dense model as supervision promotes higher alignment between its own attention patterns and the text supervision. It improves performance compared to potentially inconsistent human summaries, as shown in the last two rows of Table 2.

## 6 EXPERIMENT

### 6.1 SETUPS

We brief the experiment setups here, with more details in Appendix A.

**Baselines**. We compare MoA with state-of-the-art static and dynamic sparse attention methods, including StreamingLLM (Xiao et al., 2024c), InfLLM (Xiao et al., 2024a) and H2O (Zhang et al., 2023). We define the *density* of an LLM as the ratio of the average in-memory KV-Cache length to the sequence length during the sparse decode stage. Notably, in MoA and StreamingLLM, KV-Cache length equals the attention span during the sparse prefill stage. In contrast, H2O use dense prefill. Besides, H2O and InfLLM require additional computations to dynamically determine the KV-Cache.

**Models and Benchmarks**. We evaluate on vicuna-{7b, 13b}-v1.5-16k models (Chiang et al., 2023) from LMSys and Llama-3-{8b, 70b}-Instruct-262k models (AI, 2024) from Gradient AI. For long-context retrieval, we use LongEval (Li et al., 2023a) to test key-value retrieval accuracy with 100 data items per length level. For long-context understanding, we use LV-Eval (Yuan et al., 2024) and LongBench (Bai et al., 2023), which include 11 and 13 sub-datasets, respectively. For coherence testing, we measure perplexity on four long datasets (Dasigi et al., 2021; Fabbri et al., 2019; Li & Roth, 2002; Hovy et al., 2001; Mohler et al., 2016) with diverse tasks. Unless otherwise specified,

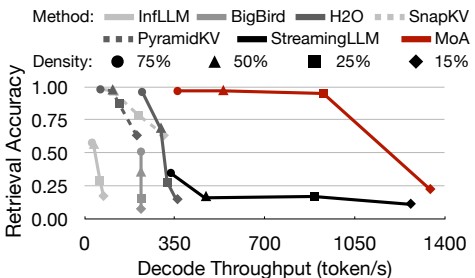

Figure 4: Accuracy-throughput trade-offs of seven attention methods at different densities, tested on Vicuna-7B with 8k input length using one A100-80GB GPU on the LongEval dataset.

| Mask Design | Retrieval Acc. | | PPL | |
|---|---|---|---|---|
| | 8k | 16k | 8k | 12k |
| Uniform | 0.25 | 0.15 | 4.89 | 5.19 |
| +Hetero. Layers | 0.31 | 0.26 | 4.55 | 4.85 |
| +Hetero. Heads | 0.95 | 0.41 | **3.96** | 4.30 |
| +Elastic | **0.98** | **0.43** | **3.96** | **4.29** |

Table 3: Ablation study on search space with consistent 25% density, progressively introducing heterogeneity in layers, heads, and elastic rules. Evaluations are done with retrieval accuracy and perplexity.

performance experiments are restricted to eight A100-80GB GPUs over a 24-hour period, with OOM (Out-Of-Memory) and OOT (Out-Of-Time) conditions noted. Efficiency experiments measure the decode throughput on a single A100-80GB GPU at maximum batch sizes of respective methods.

**MoA Settings**. We restrict the number of distinct rules to at most two per model layer to ensure inference-time efficiency. We profile MoA on MultiNews (Fabbri et al., 2019) with model summaries at 2k, 4k, and 8k lengths. The optimal compression plan is selected with the validation dataset at 12k. Each model uses the same plan across all benchmarks and lengths. The models are not fine-tuned.

### 6.2 Accuracy-Throughput Trade-off

Figure 4 shows that MoA advances the Pareto Front in context retrieval accuracy and decode throughput across varied densities and six baselines. At the same densities, MoA notably enhances throughput by $1.6\times$ to $18.1\times$ compared to H2O, InfLLM, BigBird (Zaheer et al., 2020), SnapKV (Li et al., 2024c) and PyramidKV (Cai et al., 2024), due to its efficient static attention design. The throughput even outperforms StreamingLLM thanks to our customized GPU kernel. Additionally, MoA achieves notably higher retrieval accuracies across a range of densities. We conduct extensive evaluations for MoA 's performance and efficiency on various benchmarks across context lengths from 4k to 256k and model sizes ranging from 7B to 70B in subsequent sections.

### 6.3 Performance

MoA outperforms state-of-the-art sparse attention methods across various model sizes and benchmarks, achieving comparable performance to the original dense model at 50% density.

**Long-Context Retrieval**. As shown in Table 4, MoA demonstrates a maximum of 8% relative accuracy drop (calculated as $\max\{1 - \text{Acc.}_{\text{MoA}}/\text{Acc.}_{\text{Original}}\}$ across three lengths and LLMs), significantly less than the 87%, 58% and 44% for StreamingLLM, InfLLM and H2O. On average, the relative accuracy drop for MoA is under 1%, much less than others at 51%, 41% and 20%, respectively. Figure 5(a) shows that MoA retains over 90% retrieval accuracy up to 60k lengths, equaling the dense model's effective context length. Note that it is done within 8k profiling and 12k validation. In contrast, the effective context lengths for H2O, InfLLM, and StreamingLLM are only 8k, <4k, and <4k, respectively. Appendix B.1.2 shows that MoA extends its effective context to approximately $3.9\times$ the average KV-Cache length.

**Long-Context Understanding**. As shown in Table 4, MoA minimizes the maximum relative performance drop in LV-Eval and LongBench benchmarks to only 5% and 3%, respectively—much lower than the 36% and 18% experienced by StreamingLLM. H2O and InfLLM show maximum relative drops of 9%-17% and 3%-5% with higher efficiency costs. Similar trends show in perplexity tests, where MoA maintains less than 1% relative perplexity increase, while others exhibit 4%-13% increases. This trend holds for other densities, as shown in Appendices B.1.1 and B.1.3. Figure 10 and Table 8 further details the score with different tasks. MoA achieves comprehensive performance comparable to the original dense model, as well as H2O that requires higher efficiency cost. In contrast, StreamingLLM and InfLLM display inconsistent performance: it sometimes surpasses the original model in some tasks, while suffering noticeable degradation in others.

Table 4: Comparative analysis of retrieval accuracy, LV-Eval scores, LongBench scores, and perplexity for various models with different attention methods. All sparse methods employ 50% density in decode stage. H2O uses dense prefill, while StreamingLLM, InfLLM and MoA use sparse prefill. InfLLM for 70B model is excluded due to OOT issues.

| Model | Attention | Retrieve Acc. ↑ | | | LV-Eval ↑ | LongBench ↑ | PPL ↓ |
| | | 4k | 8k | 16k | 16k | 0-16k | 8-12k |
|---|---|---|---|---|---|---|---|
| Vicuna-7B | Original | 1.00 | 0.98 | 0.62 | 5.93 | 34.76 | 3.79 |
| | H2O | 0.86 | 0.68 | 0.35 | 5.42 | 33.59 | 3.94 |
| | InfLLM | 0.67 | 0.57 | 0.26 | 5.13 | 32.97 | 4.07 |
| | StreamingLLM | 0.43 | 0.16 | 0.08 | 4.72 | 31.84 | 4.48 |
| | MoA | **1.00** | **0.97** | **0.57** | **5.61** | **33.96** | **3.75** |
| Vicuna-13B | Original | 0.99 | 0.98 | 0.44 | 5.83 | 39.23 | 3.62 |
| | H2O | 0.88 | 0.76 | 0.28 | 5.66 | 38.13 | 3.80 |
| | InfLLM | 0.70 | 0.53 | 0.27 | 6.80 | 37.13 | 4.07 |
| | StreamingLLM | 0.65 | 0.49 | 0.33 | 5.43 | 32.13 | 4.10 |
| | MoA | **0.99** | **0.93** | **0.49** | **7.16** | **38.77** | **3.62** |
| Llama3-8B | Original | 0.99 | 0.99 | 0.97 | 17.49 | 43.69 | 4.52 |
| | H2O | 0.94 | 0.89 | 0.88 | 16.03 | **42.99** | 4.63 |
| | InfLLM | 0.65 | 0.59 | 0.37 | 14.44 | 42.43 | 4.68 |
| | StreamingLLM | 0.68 | 0.55 | 0.52 | 11.16 | 38.22 | 4.79 |
| | MoA | **0.99** | **1.00** | **1.00** | **17.46** | 42.97 | **4.49** |
| Llama3-70B | Original | 1.00 | 0.99 | 0.93 | 24.51 | 49.10 | 3.67 |
| | H2O | 0.93 | 0.91 | OOM | OOM | OOM | OOM |
| | StreamingLLM | 0.20 | 0.15 | 0.04 | 17.45 | 42.53 | 4.26 |
| | MoA | **1.00** | **1.00** | **0.94** | **23.65** | **47.79** | **3.75** |

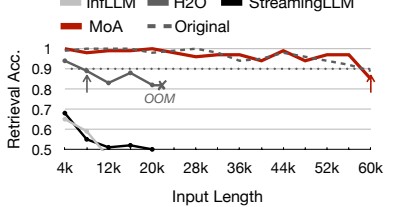

(a) Retrieval accuracy and the effective context length (arrow).

| Attention | Retrieve Acc. ↑ | | | | LV-Eval ↑ | | |
| | 32k | 64k | 128k | 256k | 32k | 64k | 128k |
|---|---|---|---|---|---|---|---|
| Original | 0.98 | 0.93 | 0.76 | 0.37 | 16.74 | 15.39 | 14.71 |
| InfLLM | 0.43 | 0.32 | 0.25 | OOT | 14.22 | 12.17 | OOT |
| StreamingLLM | 0.52 | 0.48 | 0.41 | 0.25 | 12.38 | 11.45 | 11.94 |
| MoA | **1.00** | **0.92** | **0.83** | **0.46** | **17.07** | **15.13** | **14.14** |

(b) Retrieval accuracy and LV-Eval score at longer lengths

Figure 5: Comparative analysis at extended sequence lengths with different attention methods using Llama3-8B model. All methods employ 50% density in both prefill and decode stages.

**Longer-Context Generalization**. By compressing within 12k lengths, MoA effectively generalizes to lengths of 32k-256k, as shown in Figure 5(b). At the extended lengths, MoA outperforms both InfLLM and StreamingLLM by $1.9 - 3.3\times$ in retrieval accuracy and $1.2 - 1.4\times$ in LV-Eval scores, demonstrating comparable performance to the original dense model.

**Ablation Study**. We evaluate the performance impact of different sparse mask search spaces in Table 3. Starting with a basic uniform mask, we observe significant enhancements by sequentially introducing heterogeneity: layers first, then heads, and finally elastic rules.

## 6.4 EFFICIENCY

MoA shows high runtime efficiency with a manageable one-time compression overhead.

**Runtime Efficiency**. Table 5 compares the runtime efficiency of MoA over various attention methods and LLM frameworks, with the ablation of efficiency improvements brought by each design factor of MoA. At 50% density, MoA boosts the decode throughput by $6.6\times$ to $8.2\times$ compared to FlashAttention2. It outperforms H2O and InfLLM with $1.2\times$ to $4.0\times$ decode throughput improvements. Even

Table 5: Runtime efficiency of different methods on Vicuna-7B and 13B models. Efficiency improvements of MoA are ablated with four factors. All sparse attention methods use 50% density. Decode throughput (tokens per second) evaluated at the maximum batch capacity of an A100-80GB GPU.

| Model | Framework | Attention | 4k | | 8k | | 16k | |
|-------|-----------|-----------|-------|------------|-------|------------|-------|------------|
| | | | Batch | Throughput | Batch | Throughput | Batch | Throughput |
| 7B | vLLM | PagedAttention | 30 | 628.8 | 15 | 323.0 | 8 | 145.5 |
| | FlexGen | H2O | 20 | 754.9 | 6 | 296.3 | 1 | 51.7 |
| | HuggingFace | InfLLM | 15 | 62.0 | 10 | 37.5 | 6 | 19.2 |
| | HuggingFace | StreamingLLM | 50 | 945.1 | 25 | 467.3 | 12 | 232.0 |
| | | FlashAttention2 | 30 | 134.6 | 15 | 66.9 | 8 | 32.9 |
| | | +Static KV-Cache | 30 | 496.1 | 15 | 219.5 | 8 | 91.6 |
| | HuggingFace | +Reduced Attention | 30 | 722.5 | 15 | 369.9 | 8 | 178.3 |
| | | +Increased Batch | 50 | 897.7 | 25 | 436.7 | 12 | 206.4 |
| | | +Kernel (=**MoA**) | 50 | **1099.0** | 25 | **535.7** | 12 | **257.3** |
| 13B | vLLM | PagedAttention | 16 | 314.8 | 8 | 160.5 | 4 | 71.1 |
| | FlexGen | H2O | 12 | 330.2 | 4 | 138.2 | 1 | 37.4 |
| | HuggingFace | InfLLM | 8 | 30.3 | 5 | 17.63 | 3 | 11.3 |
| | HuggingFace | StreamingLLM | 28 | 478.4 | 14 | 241.2 | 7 | 116.5 |
| | | FlashAttention2 | 16 | 81.3 | 8 | 40.8 | 4 | 19.8 |
| | | +Static KV-Cache | 16 | 264.6 | 8 | 111.3 | 4 | 62.2 |
| | HuggingFace | +Reduced Attention | 16 | 329.6 | 8 | 156.4 | 4 | 87.3 |
| | | +Increased Batch | 28 | 471.5 | 14 | 222.6 | 7 | 108.3 |
| | | +Kernel (=**MoA**) | 28 | **550.9** | 14 | **267.6** | 7 | **132.3** |

compared to the highly system-level optimized vLLM framework (Kwon et al., 2023), MoA still achieves a $1.7\times$ to $1.9\times$ throughput increase. MoA also reduces total GPU memory by $1.2\times$ to $1.4\times$, as detailed in Appendix B.2.1. Results at a 128k length are in Appendix B.2.2. This throughput gain results from four main factors: static-sized KV-Cache during generation($\approx 3.0\times$); reduced attention computations due to sparsity ($\approx 1.5\times$); increased batch sizes enabled by smaller KV-Cache memory ($\approx 1.4\times$); and our CUDA-implemented GPU kernel for MoA heterogeneous attention ($\approx 1.2\times$).

**Compression Pipeline Efficiency**. MoA completes the automatic compression pipeline for the Vicuna-7B and 13B models within two hours. For the larger Llama3-70B model, the process requires 8.5 hours of real-time and 34.7 hours of GPU time. See Appendix B.2.3. for more details.

## 6.5 RULES DISCOVERED BY MoA

We investigate MoA's elastic rules for each head. As shown in Figure 11, masks in the initial and middle layers exhibit high density, aligning with the conclusions from previous research on LLM's intrinsic dimensions (Valeriani et al., 2023) and layer sensitivities (Yuan et al., 2023). Conversely, in the final layers, most heads require low density, while few need high density. Figure 12 shows that layers with lower average density typically display more diverse densities among heads, confirming the need for heterogeneity within the same layer. Further details and insights are in Appendix C.

## 7 CONCLUSION AND FUTURE WORK

MoA automates the selection of heterogeneous elastic masks for each attention head and input length, significantly extending the effective context length of LLMs by $3.9\times$. It enhances retrieval accuracy by $1.5\times$ to $7.1\times$ over uniform sparse attention method and increases throughput to over $7\times$ at 50% average density, maintaining performance on par with dense models in rigorous benchmarks.

**Limitations and Future Work**. Under an extremely low-density budget, MoA fails to maintain good performance. Designing a dynamic MoA method has the potential to address this issue, which we leave for future work. Using non-linear elastic rules with bounded attention spans is also worth exploring. Additionally, MoA's profiling method can be adapted to evaluate the influence of weights and other activations, facilitating other compression methods such as quantization.

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

# A   DETAILED EXPERIMENT SETUP

## A.1   MAIN SETUP

**Baselines**. In the setup for our experiment, we adhere to specific configurations outlined in the respective papers. In the case of StreamingLLM (Xiao et al., 2024c), the initial four tokens remain unmasked, serving as the attention sink, except for the 70b model in Table 4 and the super long setting in Figure 5, where we use 64 tokens as the attention sink. For InfLLM (Xiao et al., 2024a), we adhere to the original configuration by maintaining the same local window size and selected memory size, using 128 initial tokens as specified in their setup. For H2O (Zhang et al., 2023), we ensure the same number of heavy hitter tokens and recent tokens. Note that H2O uses dense prefill since it relies on the column sum of the attention matrix to calculate the importance of every token for KV-Cache eviction. StreamingLLM, InfLLM and MoA use sparse prefill.

**Models and Benchmarks**. Since vicuna-7b-v1.5-16k and vicuna-13b-v1.5-16k (Chiang et al., 2023) can only take in 16k context length, we use the 16k split of LV-Eval benchmark (Yuan et al., 2024), truncating the input to 15500 for model input in Table 4. For the LongBench benchmark (Bai et al., 2023), we use the LongBench-E split, which features a balanced number of data items at every length level. The LongBench dataset is segmented into ranges of 0-4k, 4-8k, and 8k+ tokens. We test each split using the input length truncation thresholds of 3,500, 7,500, and 15,500 tokens, respectively.

**Perplexity Evaluation**. We construct a comprehensive yet concise test set by sampling $50 \times 4$ data items for each length level from the test split of four long-context understanding datasets: Qasper (Dasigi et al., 2021), MultiNew  (Fabbri et al., 2019), TREC (Li & Roth, 2002; Hovy et al., 2001) and LCC (Mohler et al., 2016), representing the question answering, summarization, few-shot learning, and code completion abilities of the LLM. Following LongBench, the data items are organized as question-answer pairs. The questions and answers are written by humans and come with the dataset. The perplexity is calculated solely on the answer part of the data, demonstrating the model's coherence in responding to user requests.

**Validation Dataset**. The validation dataset is used to select the optimal compression plan among the Pareto front solutions during the optimization step. The validation dataset is similarly constructed as the perplexity test dataset, but on the respective validation split of the datasets. $50 \times 4$ data items are sampled from the same four long-context understanding datasets: Qasper (Dasigi et al., 2021), MultiNew  (Fabbri et al., 2019), TREC (Li & Roth, 2002; Hovy et al., 2001) and LCC (Mohler et al., 2016). The additional $50$ data items from the LongEval (Li et al., 2023a) dataset are also added to validate the retrieval ability. For the datasets that do not contain the validation split, namely TREC, MultiNews and LCC, we sample from the test split and ensure different data items with the perplexity evaluation dataset.

**MoA Settings**. MoA uses the block sparse attention pattern with a block size of 64, where each grid depicted in Figure 3(a) represents a block. The first block of tokens is not masked as the attention sink. For the profile stage, we use the MultiNews (Fabbri et al., 2019) calibration dataset with model response as supervision, as described in Section 5. We use $50 \times 3$ data items at 2k, 4k, 8k lengths. The data items are padded to their corresponding length level in order to ensure a unified shape of attention influence tensors for each length level. We adopt block granularity during the profiling stage, calculating the average attention influence within each block to represent the block's overall influence. For hyperparameter search space $\alpha$ and $\beta$, we use 6 values for $\alpha$ and 9 values for $\beta$, creating a search space of 54 pairs for each attention head. $\alpha$ is uniformly sampled from the range $[-2048, 8192]$, and $\beta$ is uniformly sampled from $[0, 1]$. The resulting attention span lengths are clipped to the range between 0 and the current input length. The optimization is done with the multi-objective optimization at the same set of lengths. We limit the number of distinct rules to at most two per model layer to ensure inference-time efficiency. Among the Pareto front solutions, we select the one with the lowest perplexity on the validation dataset of length 12k.

## A.2   EFFICIENCY EXPERIMENT SETUP

We test the efficiency of different frameworks using a single NVIDIA A100-SXM4-80GB GPU. To improve the runtime profiling accuracy, we first run five forward passes as warmups. Then we use `torch.CudaEvent` to calculate the runtime for each method. Our experiments are structured around three scenarios: including prefilling 3k tokens and decoding 1k tokens; prefilling 6k tokens

and decoding 2k tokens; prefilling 12k tokens and decoding 4k tokens. The labels are marked by the total sequence length, which equals prefill length plus decode length.

For MoA, The implementation is based on Huggingface Transformers. During the prefill stage, we use the sparse CUDA kernel designed by us with block size 64. During the decode stage, we modify the KV-Cache implementation to support our heterogeneous elastic rules. Thanks to our fixed sliding-window span during the decode stage, we simply replace the old KV-Cache that exceeds the span with the latest KV-Cache. Our custom decoding CUDA kernel then handles KV-Cache with varying lengths across different attention heads during the decoding process.

For H2O, we use its official efficient implementation, which is based on Flexgen (Sheng et al., 2023). Note that H2O uses dense prefill since it relies on the column sum of the attention matrix to calculate the importance of every token for KV-Cache eviction, which requires the attention matrix to be explicitly calculated. It makes H2O's prefill stage currently incompatible with kernel optimizations like FlashAttention. Therefore, H2O is easy to get OOM (Out-Of-Memory) with large prefill length and increased batch size.

In our efficiency tests across all frameworks, we implemented a simple optimization at the language modeling head (lm head) during the prefill stage. Specifically, after the final layer of the transformers, we compute the logits—these are the raw outputs that are transformed into probabilities—for only the last token. This selective computation avoids generating these probabilities for preceding tokens, substantially reducing both computational overhead and memory usage. We also set the environment variable `PYTORCH_CUDA_ALLOC_CONF` to be `expandable_segments:True` for Hugginface and MoA to mitigate memory fragmentation, allowing larger inference batch size.

Following the performance experiments, we use Vicuna-7B and Vicuna-13B for efficiency tests whenever possible. However, the official efficient implementation of H2O based on Flexgen only supports OPT (Zhang et al., 2022). Therefore, we use OPT-6.7b and OPT-13b models for H2O in Table 11 for comparison.

### A.3 ABLATION STUDY SETUP

In the ablation study in Table 2 and Table 3, we use 25% density instead of the 50% used in the main experiment in Table 4. This decision is based on the observation that at a density of 50%, the performance of the various designs is quite similar, making it difficult to discern significant differences. In contrast, a lower density of 25% reveals more pronounced disparities between the designs, providing a clearer basis for comparison.

In the calibration dataset experiments in Table 2, we intentionally exclude the influence of the validation dataset. We avoid using the validation dataset by profile and optimize solely at 8k length, reducing the multi-objective optimization problem to a single-objective one with only one optimal compression plan instead of a set of Pareto fronts.

### A.4 INPUT FORMAT AND EXAMPLES

We list the prompt format and input examples used in our primary experiments and datasets. Dashed lines are included only for illustration clarity and are not part of the texts given to the LLMs.

---

Format 1. **LongEval**

Below is a record of lines I want you to remember. Each line begins with 'line <line index>' and contains a '<REGISTER_CONTENT>' at the end of the line as a numerical value. For each line index, memorize its corresponding <REGISTER_CONTENT>. At the end of the record, I will ask you to retrieve the corresponding <REGISTER_CONTENT> of a certain line index. Now the record start:

- - - - - - - - - - - - - - - - - - - - - - - - - - - - - - - - - - - - - - - - - - - -

line delightful-incandescence: REGISTER_CONTENT is <19147>
line **cloistered-presence**: REGISTER_CONTENT is **<8862>**
...

- - - - - - - - - - - - - - - - - - - - - - - - - - - - - - - - - - - - - - - - - - - -

Now the record is over. Tell me what is the <REGISTER_CONTENT> in line **cloistered-presence**? I need the number.

---

Format 1 illustrates the input format for the LongEval (Li et al., 2023a) retrieval benchmark. The instruction indicating which line to retrieve is provided after a lengthy context containing massive lines of register contents to remember.

---

Format 2. **Needle-In-A-Haystack (NIAH)**

People who are powerful but uncharismatic will tend to be disliked. Their power makes them a target for criticism that they don't have the charisma to disarm. That was Hillary Clinton's problem.
**The best thing to do in San Francisco is eat a sandwich and sit in Dolores Park on a sunny day.**
It also tends to be a problem for any CEO who is more of a builder than a schmoozer.
...

- - - - - - - - - - - - - - - - - - - - - - - - - - - - - - - - - - - - - - - - - - - -

What is the best thing to do in San Francisco?

---

Format 2 depicts the input format for another common retrieval benchmark, Needle-In-A-Haystack (NIAH) (Kamradt, 2024). The NIAH test comprises a single "needle" sentence that commonly does not fit into an irrelevant context. The model tries to answer the question based on this needle sentence.

---

Format 3. **MultiNews Calibration Dataset**

You are given several news passages. Write a one-page summary of all news.
<News1>
<News2>
...
Now, write a one-page summary of all the news.

- - - - - - - - - - - - - - - - - - - - - - - - - - - - - - - - - - - - - - - - - - - -

**<Summarization>**

---

Format 3 demonstrates the input format for our calibration dataset. The long-contextual MultiNews dataset (Fabbri et al., 2019) consists of multiple news documents. The context includes a prompt instructing the original dense model to generate a summarization for these news articles, reflecting long-range dependencies and model alignment. The generated summarization serves as supervision during the cross-entropy loss calculation at the profiling stage.

Table 6: Comparative analysis of retrieval accuracy, LV-Eval scores, LongBench scores, and perplexity for various models with different attention methods. All methods employ 75% density in both prefill and decode stages.

| Model | Attention | Retrieve Acc. ↑ | | | LV-Eval ↑ | LongBench ↑ | | | PPL ↓ |
| | | 4k | 8k | 16k | 16k | 0-4k | 4-8k | 8-16k | 8-12k |
|---|---|---|---|---|---|---|---|---|---|
| Vicuna-7B | StreamingLLM | 0.91 | 0.35 | 0.09 | 4.30 | 36.39 | 32.44 | 31.04 | 3.92 |
| | MoA | **1.00** | **0.97** | **0.58** | **5.67** | **38.07** | **33.80** | **31.75** | **3.78** |
| Vicuna-13B | StreamingLLM | 0.73 | 0.81 | 0.37 | **5.65** | 36.77 | 34.65 | 33.43 | 3.70 |
| | MoA | **0.99** | **0.97** | **0.42** | 5.57 | **41.85** | **39.76** | **36.06** | **3.62** |
| Llama3-8B | StreamingLLM | **1.00** | 0.83 | 0.76 | 14.89 | 42.45 | 40.62 | 42.51 | **4.51** |
| | MoA | 0.99 | **1.00** | **0.93** | **15.61** | **43.51** | **43.16** | **43.58** | 4.53 |

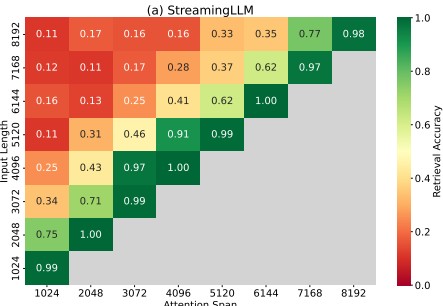 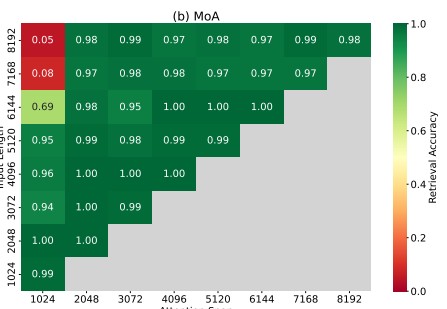

Figure 6: Retrieval accuracy of Vicuna-7B model using different attention methods across varying attention spans and input lengths. The X-axis shows different attention spans; the Y-axis shows different input lengths for the retrieval task. Subfigure (a) shows results for StreamingLLM, and subfigure (b) for MoA.

# B ADDITIONAL EXPERIMENT RESULTS

## B.1 PERFORMANCE

### B.1.1 OVERALL PERFORMANCE

Table 6 shows the overall performance of MoA at a higher density of 75%. MoA shows improved performance over the baseline with the uniform attention baseline. The progressive change of performance with respect to different densities is also shown in Figure 7(b) and Figure 9

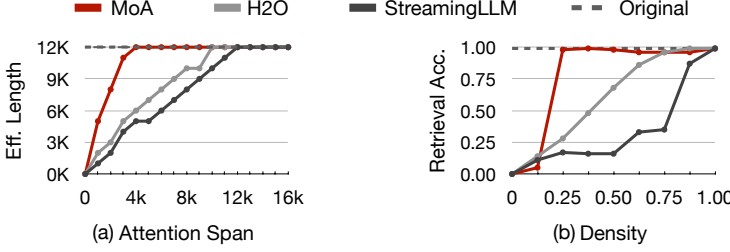

Figure 7: Retrieval accuracy tests on LongEval with Vicuna-7B. (a) Varies input lengths and densities to show effective context lengths across attention spans, (b) Set input length at 8k and show retrieval accuracy across different densities.

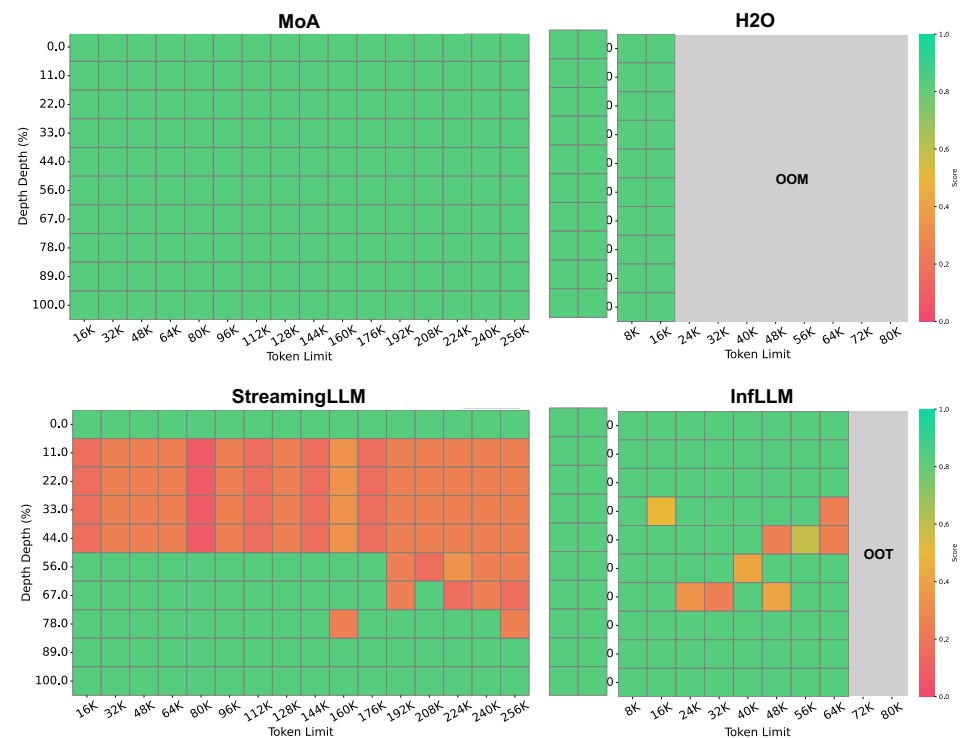

Figure 8: The Needle-In-A-Haystack (NIAH) retrieval accuracy using different attention methods across 8k to 256k input lengths on Llama-3-8B model. All sparse attention methods employ a 50% density.

### B.1.2 LONG-CONTEXT RETRIEVAL

**LongEval Retrieval**. We conduct a detailed experiment to test the retrieval ability of different attention methods across various attention spans and input lengths with the LongEval (Li et al., 2023a) dataset.

Figure 6 shows the detailed data for effective context length calculation. As shown in the figure, StreamingLLM can hardly maintain retrieval accuracy when the input length is beyond the attention span, while MoA can effectively extend the effective context length.

Following previous work (Chen et al., 2023; Tworkowski et al., 2023), we quantify effective context length as the maximum input length where retrieval accuracy remains above a 90% threshold. As shown in Figure 7(a), StreamingLLM and H2O achieve effective context lengths of no more than 2k tokens beyond their attention spans. In contrast, MoA expands its effective context length to approximately $3.9\times$ its attention span before reaching up to the 12k limit of the original model. Figure 7(b) further shows that at a fixed input length of 8k, MoA reaches over 0.9 retrieval accuracy with just 25% density, whereas StreamingLLM and H2O require 100% and 75% density, respectively.

**Needle-In-A-Haystack (NIAH) Retrieval**. We also conduct the retrieval task using the Needle-In-A-Haystack (NIAH) dataset (Kamradt, 2024). As shown in Figure 8, MoA achieves perfect retrieval accuracy across input lengths ranging from 8k to 256k. In comparison, StreamingLLM demonstrates a limited effective context length, while InfLLM exhibits reduced retrieval accuracy within 64k input lengths. Notably, H2O and InfLLM are unable to complete tests at extreme lengths due to Out-Of-Memory and Out-Of-Time errors. These findings align with the results observed in the LongEval benchmark throughout the paper.

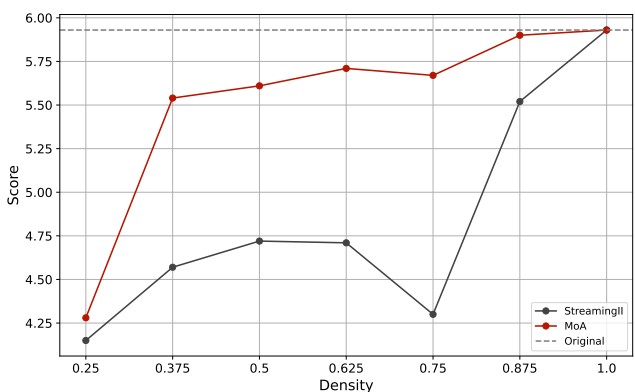

Figure 9: LV-Eval score of StreamingLLM and MoA at various densities on Vicuna-7B model.

Table 7: LongBench scores for various models with different attention methods. All methods employ 50% density in the decode stage.

| Model | Attention | LongBench ↑ | | |
| | | 0-4k | 4-8k | 8-16k |
| --- | --- | --- | --- | --- |
| Vicuna-7B | Original | 37.91 | 33.82 | 32.54 |
| | H2O | 36.23 | 32.74 | 31.81 |
| | InfLLM | 35.23 | 33.54 | 30.15 |
| | StreamingLLM | 30.53 | 33.28 | 31.70 |
| | MoA | 37.04 | 32.90 | 31.94 |
| Vicuna-13B | Original | 42.25 | 39.52 | 35.93 |
| | H2O | 41.63 | 38.02 | 34.75 |
| | InfLLM | 39.36 | 37.66 | 34.36 |
| | StreamingLLM | 30.65 | 33.07 | 32.68 |
| | MoA | 41.73 | 38.88 | 35.69 |
| Llama3-8B | Original | 44.27 | 43.53 | 43.26 |
| | H2O | 43.46 | 43.01 | 42.50 |
| | InfLLM | 42.78 | 42.69 | 41.81 |
| | StreamingLLM | 37.20 | 38.02 | 39.43 |
| | MoA | 43.07 | 42.75 | 43.09 |
| Llama3-70B | Original | 50.70 | 48.05 | 48.55 |
| | H2O | 50.16 | 47.77 | OOM |
| | StreamingLLM | 45.14 | 42.40 | 40.04 |
| | MoA | 49.74 | 46.80 | 46.84 |

### B.1.3 LONG-CONTEXT UNDERSTANDING

We conduct experiments with various densities on the LV-Eval benchmark (Yuan et al., 2024). As shown in Figure 9, MoA constantly outperforms the uniform static attention baseline StreamingLLM at various densities, demonstrating the effectiveness of our heterogeneous elastic rules.

We detailed the respective scores for LongBench and LV-Eval in Table 7 and Table 8. The number in the bracket of Table 8 indicates the number of sub-datasets for the category.

Table 8: Performance comparison across different models and attention methods with the LV-Eval dataset. The numbers in brackets indicate the number of sub-datasets for the category.

| Model | Attention | Single-QA | | Multi-QA | | Retrieval |
| | | w/o. Conf (2) | w. Conf (2) | w/o. Conf (3) | w. Conf (2) | w. Conf (2) |
|---|---|---|---|---|---|---|
| Vicuna-7B | Original | 10.49 | 6.29 | 6.83 | 5.60 | 0.00 |
| | H20 | 9.16 | 6.20 | 6.44 | 4.80 | 0.00 |
| | InfLLM | 7.11 | 6.70 | 6.07 | 4.80 | 0.00 |
| | StreamingLLM | 7.54 | 5.90 | 5.98 | 3.56 | 0.00 |
| | MoA | 9.98 | 6.27 | 6.16 | 5.31 | 0.09 |
| Vicuna-13B | Original | 10.64 | 7.28 | 5.32 | 5.07 | 1.08 |
| | H20 | 9.53 | 6.54 | 5.25 | 5.36 | 1.83 |
| | InfLLM | 10.21 | 9.35 | 6.03 | 3.19 | 2.08 |
| | StreamingLLM | 9.05 | 5.86 | 5.37 | 3.19 | 3.70 |
| | MoA | 11.04 | 6.93 | 5.79 | 5.84 | 6.88 |
| Llama3-8B | Original | 34.05 | 19.51 | 11.41 | 17.70 | 7.84 |
| | H20 | 28.52 | 17.05 | 11.11 | 15.98 | 9.95 |
| | InfLLM | 24.94 | 17.75 | 10.61 | 14.80 | 6.04 |
| | StreamingLLM | 20.21 | 9.57 | 8.14 | 9.36 | 10.03 |
| | MoA | 32.98 | 20.53 | 10.65 | 17.57 | 8.98 |
| Llama3-70B | Original | 44.44 | 25.02 | 16.71 | 22.86 | 17.43 |
| | StreamingLLM | 26.63 | 14.22 | 14.04 | 14.70 | 19.38 |
| | MoA | 42.44 | 23.58 | 15.75 | 21.27 | 19.19 |

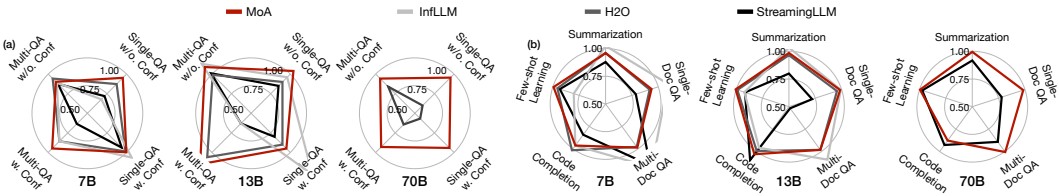

Figure 10: (a) LV-Eval and (b) LongBench scores for different attention methods at 50% density, tested on Vicuna-7B, 13B and Llama3-70B models. Scores normalized against the original dense model.

### B.1.4 LONGER-CONTEXT GENERALIZATION

We compare the retrieval accuracy with more recent works SnapKV (Li et al., 2024c) and Pyra-midKV (Cai et al., 2024) on context lengths of 32K to 256K. As shown in Table 9, MoA constantly outperforms the two latest baselines at longer contexts.

### B.1.5 INSTRUCTION-FOLLOWING GENERATION

We evaluate MoA 's performance on general instruction-following tasks using the AlpacaEval 2.0 benchmark (Li et al., 2023b; Dubois et al., 2024). Following the official setup, we compare the model's output with *gpt4_turbo* using the standard *weighted_alpaca_eval_gpt4_turbo* evaluator, which leverages the *gpt-4-1106-preview* model. The benchmark consists of inputs and outputs with average lengths of approximately 50 and 450 tokens, respectively. To accommodate the short input lengths while maintaining a density of around 50% during generation, we set the expected total token length to 512 and adjust hyperparameters across all methods accordingly.

Thanks to its elastic design, MoA employs the same compression plan used in experiments with input lengths ranging from 4k to 256k. As shown in Table 10, MoA achieves the highest length-controlled win rate, outperforming both sparse attention baselines and the original model.

Table 9: Retrieval accuracy at longer lengths for more recent baselines, tested at 50% density.

| Attention | Retrieve Acc. ↑ | | | |
|---|---|---|---|---|
| | 32k | 64k | 128k | 256k |
| SnapKV | 1.00 | 0.88 | 0.71 | 0.33 |
| PyramidKV | 1.00 | 0.85 | 0.62 | 0.37 |
| MoA | **1.00** | **0.92** | **0.83** | **0.46** |

Table 10: Length-controlled win rate and its standard error of Vicuna-7B with different attention mechanisms on AlpacaEval 2.0 benchmark. All sparse methods employ 50% density during decoding.

| Attention | Length-controlled Win Rate ↑ | Standard Error |
|---|---|---|
| Original | 8.84 | 0.53 |
| H2O | 9.66 | 0.55 |
| InfLLM | 5.76 | 0.42 |
| StreamingLLM | 7.96 | 0.49 |
| MoA | **9.83** | 0.57 |

## B.2 EFFICIENCY

### B.2.1 MEMORY AND THROUGHPUT BREAKDOWN

Table 11: Efficiency analysis of different frameworks on 7B and 13B models. H2O and MoA use 50% density. GPU memory evaluated with batch sizes 8 (7B model) and 4 (13B model).

| Size | Framework | Memory (GB) | | |
|---|---|---|---|---|
| | | 4k | 8k | 16k |
| 7B | FlashAttn2 | 28.5 | 44.4 | 76.3 |
| | H2O | 36.9 | OOM | OOM |
| | MoA | **22.7** | **32.9** | **53.5** |
| 13B | FlashAttn2 | 36.8 | 49.2 | 74.0 |
| | H2O | 40.4 | 77.9 | OOM |
| | MoA | **32.0** | **39.6** | **55.0** |

Table 11 highlights the memory efficiency of MoA compared to H2O and FlashAttention2 on 7B and 13B models. Notably, H2O runs into Out-Of-Memory (OOM) issues at longer input lengths. In contrast, MoA achieves a significant reduction in memory consumption, using $1.2$ to $1.4\times$ less memory compared to FlashAttenion2.

We further explain the decode throughput breakdown in Table 5, compared to the baseline comprising Huggingface with FlashAttention2. The observed increase in throughput primarily stems from four aspects:

**Static KV-Cache**. MoA only maintains the tokens within the span of each head, thereby preventing growth in the KV-Cache size. This strategy eliminates the need for additional memory allocation.

**Reduced Attention Computation**. MoA with features reduced density in attention span and KV-Cache. It decreases the computation and memory access required for attention computation.

**Increased Batch Size**. With the reduced size of KV-Cache, MoA supports a larger batch size, contributing to the increase in throughput.

**GPU Kernel Design**. We customize MoA GPU kernel using CUDA to support heterogeneous attention patterns with high efficiency.

Table 12: Runtime efficiency at 128k input length across different methods on Vicuna-7B and 13B models. All sparse attention methods use 50% density. Decode throughput (tokens per second) is measured with a batch size of 1, using the minimum number of A100-80GB GPUs required for testing. H2O encounters OOM error with 8 GPUs.

| Model Size | Framework | Attention | Min. #GPU | Total Throughput | Total Memory (GB) | Throughput per GPU |
|---|---|---|---|---|---|---|
| 7B | vLLM | PagedAttention | 2 | 30.2 | 142.0 | 15.1 |
| | FlexGen | H2O | >8 | - | OOM | - |
| | HuggingFace | InfLLM | 1 | 6.1 | 47.7 | 6.1 |
| | HuggingFace | StreamingLLM | 1 | 19.8 | 43.9 | 19.8 |
| | HuggingFace | FlashAttention2 | 2 | 4.3 | 85.6 | 2.2 |
| | HuggingFace | MoA | 1 | 20.3 | 44.0 | 20.3 |
| 13B | vLLM | PagedAttention | 2 | 21.5 | 142.0 | 10.8 |
| | FlexGen | H2O | >8 | - | OOM | - |
| | HuggingFace | InfLLM | 1 | 4.3 | 78.6 | 4.3 |
| | HuggingFace | StreamingLLM | 1 | 14.0 | 64.6 | 14.0 |
| | HuggingFace | FlashAttention2 | 2 | 3.0 | 130.6 | 1.5 |
| | HuggingFace | MoA | 1 | 14.7 | 63.4 | 14.7 |

### B.2.2 EFFICIENCY RESULTS FOR LONGER INPUT

We evaluate the runtime efficiency of Vicuna-7B and 13B models at a 128k input length with a single batch size. Thanks to the reduced KV-Cache, MoA efficiently processes 128k input using only one A100 GPU, whereas FlashAttention2 and vLLM baselines require at least two GPUs to handle a single request. As shown in Table 12, MoA achieves a $4.7\text{-}4.9\times$ decode speedup compared to FlashAttention2, while using half the number of GPUs. Additionally, it demonstrates a $1.9\text{-}2.1\times$ reduction in GPU memory usage. Compared to vLLM, which utilizes tensor parallelism, MoA delivers $1.3\text{-}1.4\times$ higher throughput per GPU, alongside significant memory savings.

### B.2.3 AUTOMATIC COMPRESSION PIPELINE OVERHEAD

Table 13: Compression overhead for various stages of MoA across models with differing parameter sizes, reported as the amount of GPU $\times$ latency, except when only one GPU is used. Larger models necessitate more GPUs due to model parallelism. All stages utilize GPUs, except for the Optimize stage, which uses the CPU.

| Stage | 7B LLM | 13B LLM | 70B LLM |
|---|---|---|---|
| Calibration Data Gen. | 10min | 15min | $2 \times 60$min |
| Profile | 20min | $2 \times 25$min | $8 \times 210$min |
| Optimize (CPU) | 30min | 25min | 100min |
| Validate | 35min | 40min | $2 \times 140$min |
| Total Latency | 1h 35min | 1h 45min | 8h 30min |
| Total GPU Time | 1h 5min | 1h 45min | 34h 40min |

We present a detailed breakdown of the time usage of MoA pipeline. Table 13 summarizes the time required for various crucial phases within the MoA framework, encompassing calibration dataset generation, profiling, optimization, and validation, on the Vicuna-13B model.

Profiling is the most resource-demanding part of our pipeline. For a 13b model with an 8k profile length, two A100 GPUs are required. In other cases, we only need one single GPU. Profiling on a 13b model with an 8k profile length and 50 data items takes 15 minutes. Profiling on 4k and 2k lengths takes less than 5 minutes each.

Table 14: Progressive compression overhead for various stages of MoA with respect to different parameter sizes and calibration (validation) dataset sizes.

| Stage | Complexity w.r.t parameter size | Complexity w.r.t dataset size |
|---|---|---|
| Calibration Dataset Gen. | Linear | Linear |
| Profile | Linear | Linear |
| Optimize | Polynomial $\sim$ Exponential for #Head | Irrelevant |
| Validate | Linear | Linear |
| Empirical Latency | Almost Linear | Linear |

On the Intel(R) Xeon(R) Platinum 8358 2.60 GHz CPU, the optimization concludes within approximately 25 minutes. Typically, this phase generates around 10 compression plans. Validating each one of the compression plans takes about 4 minutes, totaling around 40 minutes.

We also show the progressive compression overhead for MoA in Table 14.

## B.3  ABLATION STUDY

### B.3.1  CALIBRATION DATASET

Table 15: Performance comparison on various test sets, using different calibration sets. Tested on Vicuna-7B model. The result is tested with 50% density MoA on LongBench (Bai et al., 2023) 0-4k split.

| Dataset | Long Dep. & Align Model | Test Score | | | Avg. Score |
|---|---|---|---|---|---|
| | | Qasper | MultiNews | TREC | |
| Original | NA | 28.6 | 28.2 | 56.0 | 37.6 |
| RedPajama | ✗ | 20.6 (-8.0) | 19.6 (-8.6) | **66.0** (+10.0) | 35.4 (-2.2) |
| Qasper | ✓ | 25.6 (-3.0) | **27.8** (-0.4) | 55.0 (-1.0) | 36.1 (-1.5) |
| MultiNews | ✓ | **29.0** (+0.4) | 27.5 (-0.7) | 54.0 (-2.0) | **36.8** (-0.8) |
| TREC | ✓ | 27.3 (-1.3) | 27.3 (-0.9) | 55.0 (-1.0) | 36.5 (-1.1) |

In this section, we validate the robustness of our calibration dataset design principles. We select three sub-tasks and respective datasets from the LongBench benchmark, including Qasper (Dasigi et al., 2021), MultiNews (Fabbri et al., 2019), and TREC (Li & Roth, 2002; Hovy et al., 2001). We use their training set to construct the calibration dataset, and use their respective test set in LongBench to calculate the score. Following Section 5, all calibration datasets are constructed using the original model's response to the context and questions as the supervision.

As shown in Table 15, we find that as long as the calibration dataset conforms to the long-range dependency and model alignment highlighted in section 5, the specific choice of the dataset is less important. Calibration datasets with long dependency and model alignment show somewhat similar test results on various datasets. Additionally, they all show strong generalization power to test sets other than their respective calibration dataset.

In contrast, the RedPajama dataset without long-range dependency and model alignment shows large variance on various test sets. It also differs from the performance of the original dense model, which may incur unexpected behaviors after compression. Note that though all datasets exhibit long dependency, the questions in the TREC dataset can be answered without long context. The context in the TREC dataset of LongBench is the many-shot examples, each showing a short sentence and its classification result, while the question is to classify a new short sentence. Although the context helps to determine the complete set of 50 classes, the model can also directly clarify the sentence without any context based on common knowledge. It may contribute to a high score on the TREC test set with the RedPajama calibration dataset.

# C  COMPRESSION PLAN ANALYSIS

## C.1  STATISTICS ON RULES DISCOVERED BY MoA

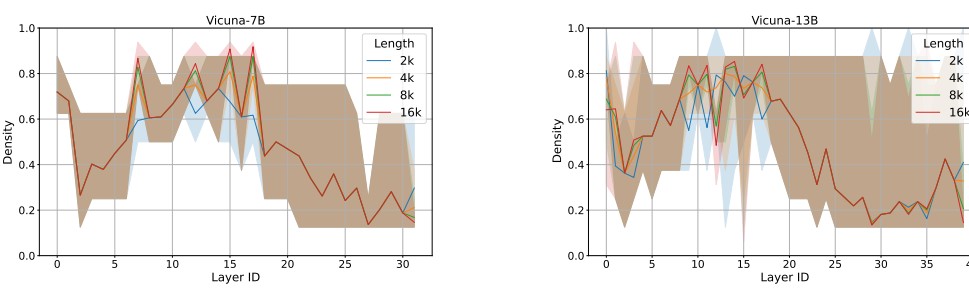

Figure 11: The MoA mask density across layers for different LLMs.

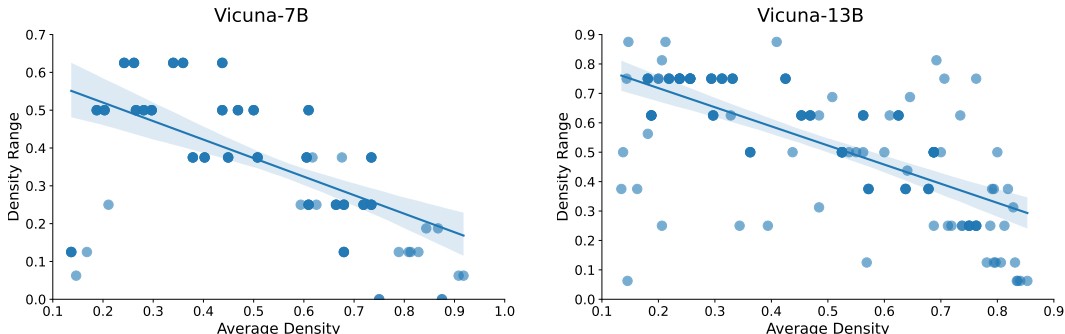

Figure 12: The MoA mask's average density and the density range for each layer for different LLMs.

This subsection provides empirical evidence for rules discovered by MoA as mentioned in Section 6.5. The lines and spans in Figure 11 show that all heads at the first few layers generally need a high KV-Cache density. Following that, a few layers generally only require medium density. Then, in the final layers, most heads require low density, while some outlier heads need high density. This observation conforms to previous findings of the intrinsic dimension of LLM Valeriani et al. (2023). The geometry of density is similar to the intrinsic dimension of LLM, with two local minima. As observed in Figure 11, layers with lower average density (smaller values on the lines) typically display a wider range of density (wider shades). Figure 12 validates such observation. This observation confirms the need for heterogeneous attention rules within the same layer.

## C.2  CONNECTIONS BETWEEN MoA RULE AND SEMANTIC

In this section, we invest the masks acquired with MoA and show the interpretable semantics of the masks. Previous works manually restrict the attention pattern of the model, which may harm the semantics learned by the dense model. In contrast, MoA preserves the semantics with statistic analysis and optimization. We use visualization, human interpretation and quantitive methods to analyze the semantics of the original model and to verify whether MoA captures such semantics.

### C.2.1  MASK VISUALIZATION AND SEMANTIC CATEGORIZATION

Given any token, two kinds of information are used as the model inputs: position encoding and token embedding. Position encoding indicates the absolute (Zhang et al., 2022) or relative positions (Touvron et al., 2023) of tokens in the sentence. Token embedding maps different tokens as different vectors. The attention head $h$ responds to both information and output the corresponding attention value $A_h$. As shown in equation 6, we denote the influence of position and token of head $h$ as function $P_h$ and $T_h$, respectively. The attention value $A_{h,i,j}$ between the $i$th and $j$th token $t_i$ and $t_j$ is determined by the combination $f_h$ of position and token influence functions.

$$A_{h,i,j} = \mathbb{A}_h(t_i, t_j, i, j) = f_h\left(P_h(i,j), T_h(t_i, t_j)\right) \qquad (6)$$

Figure 2 visualizes two typical heads that are either dominated by position $P$ or token $T$ function. For the first attention head in Figure 2, the local positional attention is clearly observed. In this head, whatever sentences are given, each token pays major attention to the first token and the prior token. As a result, the mean attention matrix accumulates extremely large attention values at the first column and the sub-diagonal. In contrast, the second attention head in Figure 2 lays more emphasis on content-based attention. Since the position distribution of important tokens are generally random, the attention matrix can show large attention values at any position. It results in a mean attention matrix without extreme mean attention values.

In conclusion, the mean attention matrix of different sentences provides a valuable insight of whether attention values of an attention head is more position-based or content-based. Intuitively, the more uneven the attention matrix value distribution is, the more position-based the head is.

### C.2.2 QUANTITATIVE SEMANTIC ANALYSIS

We quantify how much the attention head is position-based and analyze whether MoA successfully utilizes such semantics through the evaluate-generate-optimization pipeline. We model equation 6 with a linear approximation. $P_h$ and $T_h$ are random variables with the same expectation $\mu$ and standard variance $\delta$ for all heads. For attention head $h$, the weight factor $\alpha_h$ evaluates the relatively influence of position and token to the final attention value.

$$A_{h,i,j} = \alpha_h P_h(i,j) + (1 - \alpha_h) T_h(t_i, t_j) \qquad (7)$$

Given the randomness of token positions in long context, we assume that the token position and its content are irrelevant. For different sentences $s$, the expectation $\mathbb{E}_t$ of the attention value between position $i$ and $j$ can be expressed as follows. Note that it excludes the matrix diagonal since $T_h(t_i, t_j), i \neq j$ and $T_h(t_i, t_i)$ may follow different distributions.

$$\begin{aligned}
\mathbb{E}_t[A_{h,i,j}] &= \frac{1}{S} \sum_{s=1}^{S} \left( \alpha_h P_h(i,j) + (1 - \alpha_h) T_h(t_i^{(s)}, t_j^{(s)}) \right) \\
&= \alpha_h P_h(i,j) + (1 - \alpha_h) \frac{1}{S} \sum_{s=1}^{S} T_h(t_i^{(s)}, t_j^{(s)}) \\
&= \alpha_h P_h(i,j) + (1 - \alpha_h) \mu_T, \forall i > j
\end{aligned} \qquad (8)$$

The standard division $\sigma_p$ of $\mathbb{E}_t$ over different positions of the attention matrix is

$$\begin{aligned}
\sigma_p(\mathbb{E}_t[A_{h,i,j}]) &= \sqrt{\frac{2}{(1+N)N} \sum_{i,j \in [1,N), i>j} [(\alpha_h P_h(i,j) + (1-\alpha_h)\mu_T) - (\alpha_h \mu_P + (1-\alpha_h)\mu_T)]^2} \\
&= \alpha_h \delta_p
\end{aligned} \qquad (9)$$

We name $\sigma_p(\mathbb{E}_t[A_{h,i,j}])$ the Standard division of Expectation (SoE) of head $h$. Note that the expectation is taken over different sentences, while the standard division is taken over different attention positions. Since $\delta_p$ is the same for all heads, we derive that the position impact $\alpha_h$ is proportional to the SoE of different heads.

The conclusion quantifies the observation stated in Section C.2.1. Intuitively, SoE shows how uneven the mean attention matrix is, thus showing the influence of position to the attention values. MoA's generated mask density shows positive relation with SoE, suggesting that MoA successfully captures the semantic information of the dense language model as shown in Figure 13.

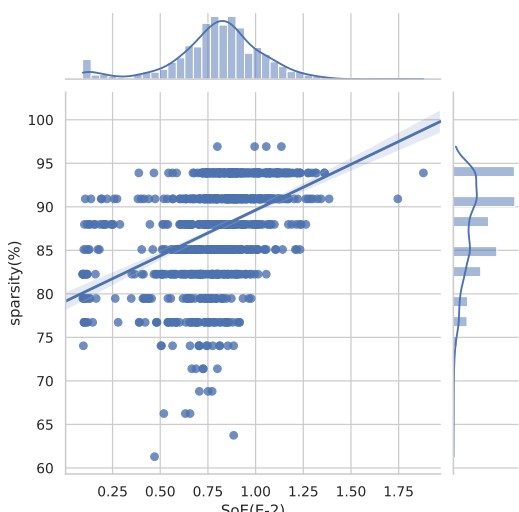

Figure 13: Positive correlation between MoA's mask sparsity and head's dependency on position (SoE).

# D AUTOMATIC PIPELINE DETAILS

## D.1 ADDTIONAL ORACLE ON ELASTIC PATTERN DESIGN

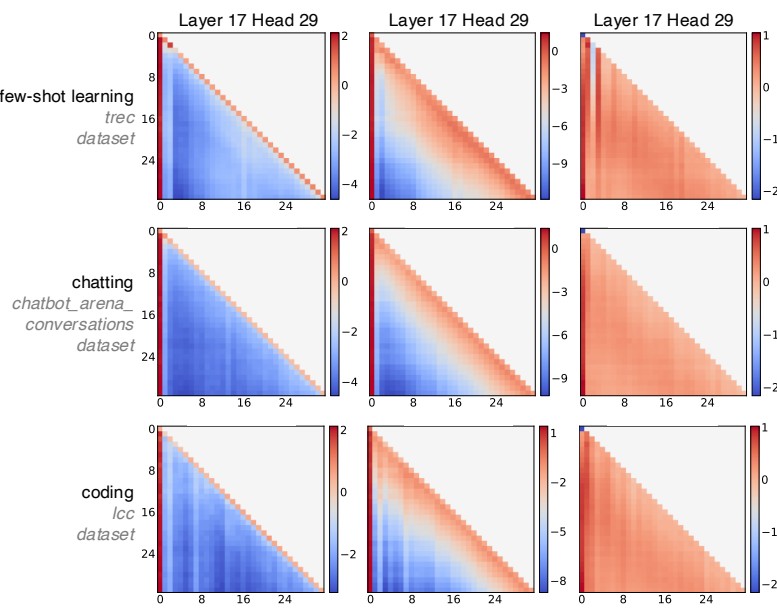

Figure 14: Examples of attention matrices from different attention heads (columns) and tasks (rows) of the Vicuna-7B model. The attention matrices were averaged over 256 data items per dataset. The same head shows a similar attention span across different tasks, explaining the robust cross-dataset generalizability of our method.

We visualize the attention matrix of the same attention heads across three additional tasks in Figure 14, as an extension of Figure 2. The consistent attention span across tasks sheds light on the strong cross-dataset generalization ability of our MoA method.

## D.2 Derivation of Attention Influence

We use the first-order Taylor expansion to calculate the influence of each attention value. This approximation approach is supported by methodologies commonly employed in other LLM compression approaches (Li et al., 2024a; Shi et al., 2021; Das et al., 2023; Jiang et al., 2023).

As discussed in Section 4.1, when masking out attention value $A_{h,i,j}$ at head $h$, row $i$, and column $j$, it also influences the attention values in the same row by $\Delta A_{h,i,n|j}$.

$$
A_{h,i,n} = \frac{e^{S_{h,i,n}}}{\sum_j e^{S_{h,i,j}}}
$$

$$
\Delta A_{h,i,n|j} = \begin{cases} -A_{h,i,n}, & n = j \\ A_{h,i,n}(\sum_j e^{S_{h,i,j}}/\sum_{j\neq n} e^{S_{h,i,j}} - 1), & n \neq j \end{cases} \tag{10}
$$

Following the definition, the attention influence $\mathbf{E}_h$ is calculated as follows:

$$
E_{h,i,j} = \sum_n \frac{\partial L}{\partial A_{h,i,n}} \cdot \Delta A_{h,i,n|j} \tag{11}
$$

Given Equation 11 and 10, we derive Equation 3 as follows. For notation simplicity, we omit the head index $h$ here.

$$
\begin{aligned}
E_{i,j} &= \sum_n \frac{\partial L}{\partial A_{i,n}} \cdot \Delta A_{i,n|j} \\
&= \frac{\partial L}{\partial A_{i,j}} \cdot (-A_{i,j}) + \sum_{n\neq j} \frac{\partial L}{\partial A_{i,n}} \cdot A_{i,n} \cdot \left( \frac{\sum_k e^{S_{i,k}}}{\sum_{k\neq j} e^{S_{i,k}}} - 1 \right) \\
&= \frac{\partial L}{\partial A_{i,j}} \cdot (-A_{i,j}) + \sum_{n\neq j} \frac{\partial L}{\partial A_{i,n}} \cdot A_{i,n} \cdot \frac{e^{S_{i,j}}}{\sum_k e^{S_{i,k}} - e^{S_{i,j}}} \\
&= \frac{\partial L}{\partial A_{i,j}} \cdot (-A_{i,j}) + \sum_{n\neq j} \frac{\partial L}{\partial A_{i,n}} \cdot A_{i,n} \cdot \frac{e^{S_{i,j}}/\sum_k e^{S_{i,k}}}{1 - e^{S_{i,j}}/\sum_k e^{S_{i,k}}} \\
&= \frac{\partial L}{\partial A_{i,j}} \cdot (-A_{i,j}) + \sum_{n\neq j} \frac{\partial L}{\partial A_{i,n}} \cdot A_{i,n} \cdot \frac{A_{i,j}}{1 - A_{i,j}} \\
&= \frac{\partial L}{\partial A_{i,j}} \cdot (-A_{i,j}) - \frac{\partial L}{\partial A_{i,j}} \cdot A_{i,j} \cdot \frac{A_{i,j}}{1 - A_{i,j}} + \sum_n \frac{\partial L}{\partial A_{i,n}} \cdot A_{i,n} \cdot \frac{A_{i,j}}{1 - A_{i,j}} \\
&= \frac{\partial L}{\partial A_{i,j}} \cdot \left( -\frac{A_{i,j}}{1 - A_{i,j}} \right) + \frac{A_{i,j}}{1 - A_{i,j}} \cdot \sum_n \frac{\partial L}{\partial A_{i,n}} \cdot A_{i,n} \\
&= -\frac{A_{i,j}}{1 - A_{i,j}} \left( \frac{\partial L}{\partial A_{i,j}} - \sum_n \frac{\partial L}{\partial A_{i,n}} \cdot A_{i,n} \right)
\end{aligned} \tag{12}
$$

It is worth noting to mention that it can also be formulated as matrix multiplications:

$$
\mathbf{E}_h = \frac{\mathbf{A}_h}{1 - \mathbf{A}_h} \cdot \left( \frac{\partial L}{\partial \mathbf{A}_h} - \left( \frac{\partial L}{\partial \mathbf{A}_h} \cdot \mathbf{A}_h \right) \mathbb{1}^{N \times N} \right). \tag{13}
$$

## D.3 Optimization Details

### D.3.1 Optimizing at Single Length

The optimization problem is formulated as follows:

$$
\arg\min \Delta L = \sum_h \Delta L_{h,r_h}, \quad \text{s.t.} \ \frac{1}{H} \sum_h d_{r_h} \leq d_{\text{constr}}. \tag{14}
$$

To transform the optimization problem into a standard Mixed-Integer Programming (MIP) framework, we introduce the binary variable $X_{h,r_h} \in \{0,1\}$. It indicates whether to select rule $r_h$ for the attention head $h$. Assume the model has $H$ attention head, and head $h$ has $R_h$ elastic rules.

$$\arg\min \frac{1}{H} \sum_{h=0}^{H-1} \sum_{r_h=0}^{R_h-1} \Delta L_{h,r_h} X_{h,r_h} \quad \text{s.\,t.} \tag{15a}$$

$$\sum_{r_h=0}^{R_h-1} X_{h,r_h} = 1, \quad h \in \{0, \cdots, H-1\} \tag{15b}$$

$$\frac{1}{H} \sum_{h=0}^{H-1} \sum_{r_h=0}^{R_h-1} d_{r_h} X_{h,r_h} \leq d_{\text{constr}} \tag{15c}$$

$$0 \leq X_{h,r_h} \leq 1, X_{h,r_h} \in \mathbb{Z}, \quad \forall h \in \{0, \cdots, H-1\}, \forall r_h \in \mathbb{R} \tag{15d}$$

In this formulation, (15a) serves as the objective function to minimize the loss, subject to the constraints that each matrix selects exactly one compression plan (15b), and the average density does not exceed $d_{\text{constr}}$ (15c). Finally, (15d) enforces that $X_{h,r_h}$ is a binary variable, indicating the selection of plans.

Additionally, to enforce the restriction that each model layer only has a limited number of different plans, we bound the norm of element-wise multiplication of $\mathbf{X}_h = \begin{bmatrix} X_{h,0} & X_{h,1} & \cdots & X_{h,R_h-1} \end{bmatrix}^\top$ in a single layer.

### D.3.2 Optimizing at Multiple Lengths

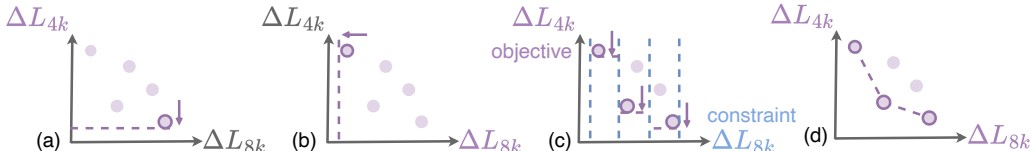

Figure 15: Illustration of our multi-objective Mixed-Integer Programming (MIP) approach, using a two-objective optimization example: (a) MoA first minimizes the loss for 4k inputs and records the corresponding loss for the current optimal plan at 8k. (b) Next, it minimizes the loss for 8k inputs and records the corresponding loss for the current optimal plan at 4k. These steps establish the loss ranges $R$ for both 4k and 8k input lengths. (c) MoA then re-optimizes the loss at 4k, this time using the loss intervals at 8k as different constraints. All plans generated under these constraints are recorded. (d) The last process (c) is repeated for 8k, using 4k intervals as constraints. Finally, plans meeting the Pareto front criteria for both 4k and 8k inputs are selected as the final outputs.

With the ability to optimize at a single length, we utilize the same framework for multi-objective MIP across various lengths. The key is to transform the multi-objective MIP problem into several single-objective MIP problems (Paria et al., 2018). We utilize the idea of epsilon-constraint method (Yv et al., 1971).

Figure 15 illustrates the optimization process for two input lengths. We discuss the generalized approach to handle an arbitrary number of lengths. We first select each input length as our primary objective to perform the single-objective optimization on it while simultaneously recording the outcomes of other objectives. Specifically, for $N$ distinct objectives, we do single-objective MIP optimization on the $i$-th objective, getting minimum loss $\Delta L_i^{(N_i)}$, and we concurrently collect losses of other objectives $\Delta L_i^{(N_j)}$ for $j \neq i$. This process allows us to establish the range of loss $R^{(N_j)} = \left[\min_i \Delta L_i^{(N_j)}, \max_i \Delta L_i^{(N_j)}\right]$ for each objective. Then, we iterate through each objective again. Compared with the original multi-objective optimization in Equation 5, we now consider other objectives as constraints. To implement this, we partition each loss range $R^{(N_j)}$ of other objectives

$j \neq i$ into $M$ uniform intervals $S_k^{(N_j)}$, where $0 \leq k < M$. We then solve the MIP problems for each objective $i$ and iterating through the constraint intervals:

$$\underset{r_h \in \mathbb{R}}{\arg\min} \, \Delta L^{(N_i)} \quad \text{s.t.} \, \frac{1}{H} \sum_{h=1}^{H} d_{r_h}^{(N_i)} \leq d_{\text{constr}}^{(N_i)}, \forall N_i \in \mathbb{N}_{\text{constr}}; \quad \Delta L^{(N_j)} \in S_{k_j}^{(N_j)}, \forall j \neq i. \quad (16)$$

where this optimization is performed for each $i$ ranging from 0 to $N$. For each $j$, $k_j$ can vary independently from 0 to $M$. For efficiency consideration, we set the number of intervals as five. Finally, the results that do not conform to the Pareto front requirements are removed, resulting in the final Pareto front set of our multi-objective optimization problem.

# E    CONCURRENT WORK

Current work RazorAttention (Tang et al., 2024a) also proposes to use heterogeneous attention for better performance. MoA distinguishes from it in the following aspects:

1. **Strategies for attention heads**: RazorAttention categorizes attention heads into two types: retrieval and non-retrieval. It adopts bipolar strategies, applying either full attention or fixed-sized local attention. In contrast, MoA recognizes the diverse attention spans of different heads and employs a broader range of strategies, covering the entire spectrum from very limited local attention to full attention.

2. **Adaptation to different input lengths**: RazorAttention uses a fixed density for non-retrieval heads across all input lengths. MoA, on the other hand, applies heterogeneous elastic rules for each head, dynamically adjusting densities based on input length while maintaining overall density constraints.

3. **Determination of strategies**: RazorAttention relies on a heuristic approach to assign strategies. It uses attention scores between the current token and specific tokens (e.g., echo and induction tokens) to identify retrieval heads and assign full attention. MoA employs a loss-based method. It uses a first-order Taylor expansion to estimate the impact of a strategy on end-to-end prediction loss and optimizes strategies to minimize this loss under density constraints.

