# OpenReview forum: "MoA: Mixture of Sparse Attention for Automatic Large Language Model Compression"
_ICLR.cc/2025/Conference — Submitted to ICLR 2025_

### Official Review · Reviewer_pLr2 · 2024-10-27

**Soundness:** 3
**Presentation:** 3
**Contribution:** 2
**Rating:** 5
**Confidence:** 3

**Summary:**

The paper introduces MoA (Mixture of Attention), a method designed to compress Large Language Models (LLMs) through the application of sparse attention mechanisms. Unlike existing methods that use a uniform sparse attention mask, MoA dynamically tailors distinct sparse attention configurations to different heads and layers, allowing it to adapt to varying input sizes and optimize for both accuracy and latency. MoA leverages long-range dependencies and model-generated summaries to enhance the calibration dataset, improving the alignment between the original dense model and the compressed version. Experiments demonstrate that MoA can significantly boost retrieval accuracy and reduce perplexity while decreasing GPU memory usage and increasing decoding throughput.

**Strengths:**

1. MoA can adjust to different input lengths, which is important for handling long sequences commonly found in real-world applications.
2.The method shows a marked improvement in retrieval accuracy and a reduction in perplexity compared to uniform attention baselines. By incorporating long-range dependencies, MoA can maintain global attention necessary for tasks requiring long-range retrieval.
3. Using model-generated summaries rather than human-written ones reduces inconsistencies and aligns attention patterns better with the original model's behavior.

**Weaknesses:**

1. The process of identifying the optimal sparse attention configuration appears to be complex, involving profiling the model, evaluating configurations, and pinpointing the best plan. This complexity may pose a barrier to practical implementation.
2. The effectiveness of MoA seems to depend heavily on the availability of appropriate long-range contextual data, such as the MultiNews dataset. The performance gains might diminish without such data. How robust the method is against different long-range contextual dataset?
3.Overall I believe this paper is good at its idea and implementation. While the topic of efficient decoding of LLMs is very hot, it is expecte to compare with more competitive related methods in the experiments to demonstrate the effectiveness of the proposed method among its competitors.

**Questions:**

See the weakness for details.

---

> ### Author Response · Authors · 2024-11-23
> **Rebuttal by Authors**
>
> ### [W1] Compression overhead seems high
>
> > The process of identifying the optimal sparse attention configuration appears to be complex, involving profiling the model, evaluating configurations, and pinpointing the best plan. This complexity may pose a barrier to practical implementation.
>
> We appreciate the reviewer's concern regarding the overhead of our compression process. We quantify the overhead of each compression step in Table 13. For the Vicuna-13B model, the entire compression pipeline can be completed in 105 minutes. Even for the 70B model, the compression can be done within 9 hours of latency and 35 GPU hours. Importantly, once compression is achieved, the model can be utilized across various tasks without any additional overhead.
>
> We also quantify the progressive overhead of our compression method for different model sizes in Table 14.
> The calibration dataset generation, profiling, and validation stages exhibit linear complexities with respect to both parameter size and dataset size.
> During the optimization stage, the complexity is determined by the number of attention heads, which typically scales with parameter size. Although this leads to an exponentially growing search space, advancements in MIP solvers allow for polynomial time solutions under many conditions [1, 2], as confirmed by our empirical results.
>
> We would also like to emphasize our efficiency design considerations during compression, particularly in handling models with extensive input lengths and numerous parameters:
>
> 1. **Elastic Attention Rules for Large Input Length:** Our method searches for length-adaptive elastic rules instead of length-dependent sparse attention masks. As demonstrated in Figure 5, by compressing within 12k input length, the same compression plan effectively generalizes up to 256k, ensuring broad applicability without re-compression.
> 2. **Efficient Optimization for Large Parameter Sizes:** We use efficient white-box optimization for the compression pipeline. During optimization, each sparsification rule for an attention head is abstracted with only two float numbers: density and accuracy loss. This abstraction greatly simplifies the problem and formalizes it as Mixed-Integer Planning (MIP), whose solving is highly optimized with existing solvers [3, 4].
>
> In conclusion, the empirical results, asymptotic analysis, strong generalization ability, and effective design choices ensure the efficiency of our automatic compression pipeline.
>
> Lastly, the compression overhead can be further reduced with system-level optimizations. Currently, we use HuggingFace model parallelism during profiling, which does not support pipelining and results in low GPU utilization. By integrating pipelining and tensor parallelism, the total GPU time can be significantly reduced in future implementations.
>
>
>
> ### [W2] Robustness for calibration dataset
>
> > The effectiveness of MoA seems to depend heavily on the availability of appropriate long-range contextual data, such as the MultiNews dataset. The performance gains might diminish without such data. How robust the method is against different long-range contextual dataset?
>
> We appreciate the reviewer’s concern regarding the robustness of MoA with respect to the calibration dataset. MoA demonstrates strong robustness as long as the calibration dataset satisfies the long-dependency and model alignment properties discussed in Section 5.
> We conduct the robustness study detailed in Appendix B.3.1. The experiments show that MoA maintains strong performance across varied calibration datasets, regardless of whether the same or different datasets are used during the compression and testing phases.
> Additionally, we explore the possible reasons for this robustness in Appendix D.1 (Figure 14). The oracle experiment on attention patterns across diverse tasks (e.g., few-shot learning, chatting, and coding) reveals that, while task-specific details may vary, the overall attention span of each head remain consistent. This consistency likely explains the strong robustness of MoA across different calibration datasets and tasks.
>
>
>
> ### [W3] Add more baselines
>
> > Overall I believe this paper is good at its idea and implementation.
> > While the topic of efficient decoding of LLMs is very hot, it is expecte to compare with more competitive related methods in the experiments to demonstrate the effectiveness of the proposed method among its competitors.
>
> We thank the reviewer and add the latest SnapKV [5] and PyramidKV [6] baselines. As shown in the updated Figure 4, MoA consistently achieves a superior throughput-accuracy trade-off across varied densities, outperforming all six baselines. Additionally, we have included the retrieval performance of     SnapKV [5] and PyramidKV [6] on longer contexts in Table 9. MoA demonstrates 4% to 21% higher retrieval accuracy compared to these baselines for 64K to 256K lengths.

---

> > ### Author Response · Authors · 2024-11-23
> > **References**
> >
> > ### **References**
> >
> > [1] Vanderbeck, F., and L. A. Wolsey. "Reformulation and Decomposition of Integer Programs.", Springer, 2010, pp. 431-502.
> >
> > [2] Korte, B., and Vygen. “Combinatorial Optimization: Theory and Algorithms“, Springer, 2012.
> >
> > [3] MOSEK ApS, 2019
> >
> > [4] Gurobi Optimization, LLC. 2024
> >
> > [5] Li, Yuhong, et al. "Snapkv: Llm knows what you are looking for before generation." arXiv preprint arXiv:2404.14469 (2024).
> >
> > [6] Cai, Zefan, et al. "Pyramidkv: Dynamic kv cache compression based on pyramidal information funneling." arXiv preprint arXiv:2406.02069 (2024).

---

> ### Author Response · Authors · 2024-11-26
> **Looking forward to your reply**
>
> Dear Reviewer pLr2,
>
> We sincerely hope our revisions address your concerns regarding **compression overhead** and **calibration dataset robustness**. We also **add two more baselines** to further demonstrate the effectiveness of MoA. We look forward to your feedback on these updates. If there are any remaining questions, please let us know at your earliest convenience. We are happy to engage in further discussions. Once again, thank you for your valuable time and contributions to refining our work.

---

> ### Comment · Reviewer_pLr2 · 2024-11-26
> **Reply to the authors**
>
> For the complexity I mentioned in weakness 1  mainly refers to the cumbersome process involved in this approach, i.e., profiling the model, evaluating configurations, and pinpointing the best plan, but not the computational overhead only. Introducing so many processes just in the sparsification of the attention modules would introduce more uncertainty in training large models, let alone mixture-structured models. And the reported compression time in your response will vary with different experimental setups. To summarize, 5 is the highest score I can give to this work.

---

> > ### Author Response · Authors · 2024-11-27
> > **Further discussions on compression pipeline complexity**
> >
> > Thank you for your valuable feedback. We understand your concern regarding the potential process complexity of MoA, and we would like to further elaborate on the practicality of our approach:
> >
> > **1. Compression Process:**
> >
> > Regarding compression complexity:
> > First, a profiling or evaluation step to analyze which module is more sensitive or which configuration is better is commonly needed for compression methods to determine the appropriate inter-module compression configurations [1,2,3].
> > Second, while our process of identifying the best plan (i.e., finding Pareto-optimal configurations and choosing the best one at an unseen long input length) may seem complex at the first glance, it is specially designed for finding an **extendable** rule to enhance **practicality**. This practicality is validated by our experiments, where we can directly apply the discovered rule at a 256k input length, even though it was derived from profiling and choosing at segment lengths of $\leq$12k, and achieve strong performance.
> >
> > Additionally, we emphasize that our compression pipeline is fully automated, requiring no manual tuning of hyperparameters, which makes it easily applicable to new models. While compression time varies with model size due to main differences in profiling time, we believe this is reasonable given that we are working with models ranging from 7B to 70B parameters. Luckily, **MoA's compression time is largely predictable based on the model size**, as most of the time is spent on the profiling step. **This predictability offers controllability of MoA's compression time**. For example, to control the overall compression time of MoA in practice, one can profile a larger model using a smaller data size.
> >
> > Finally, regarding the concern about introducing more uncertainty in training large models, we note that our entire process is performed **post-training**, with **no fine-tuning required**. This ensures that our method does not affect the model training process. We are not sure we fully understand this point, and we would appreciate it if more clarification on this point can be provided!
> >
> >
> > **2. Deployment Interface:**
> >
> > Once compression is completed, the resulting compression plan is as straightforward to deploy as its dense counterpart. It requires only a **few lines of code** to replace standard attention with our mixture-structured sparse attention. To further streamline adoption, we provide the CUDA GPU kernel for our mixture of sparse attention, integrated with the HuggingFace AttentionModule interface. The code is available via the anonymous repository link and will be open-sourced.
> >
> > By automating the compression process and offering a simple deployment interface, our approach minimizes complexity and ensures practical usability.
> >
> >
> > **References**
> >
> > [1] He, Yihui, et al. "Amc: Automl for model compression and acceleration on mobile devices." ECCV. 2018.
> >
> > [2] Dong, Zhen, et al. "Hawq: Hessian aware quantization of neural networks with mixed-precision." ICCV. 2019.
> >
> > [3] Shao, Wenqi, et al. "OmniQuant: Omnidirectionally Calibrated Quantization for Large Language Models." ICLR. 2024.

---

> > > ### Author Response · Authors · 2024-11-29
> > > **Polite reminder for response**
> > >
> > > Dear Reviewer pLr2,
> > >
> > > As the rebuttal period is nearing its conclusion, we kindly follow up to seek your response. To address any possible coding complexity concerns, we would like to highlight our anonymous repository: https://anonymous.4open.science/r/MoA-Review/README.md (also referenced in Section 1 of the submission).
> > > As demonstrated in the repository, compressing a new model is fully automated:
> > > ```
> > > python scripts/pipeline/main.py --model_path <HF_MODEL_PATH> --model_name <DEFINE_OUTPUT_DIR>
> > > ```
> > > After compression, deploying MoA is equally straightforward, requiring just two lines of code:
> > > ```
> > > model = update_model_function(model, model_name)
> > > model.model.set_mixture_of_attention(moa_config, permute_head=True)
> > > ```
> > > We hope these clarifications address your concerns. Should you have any further questions or need additional details, please feel free to contact us. We sincerely appreciate your time and effort in reviewing our work.

---

### Official Review · Reviewer_UNvV · 2024-11-03

**Soundness:** 4
**Presentation:** 2
**Contribution:** 4
**Rating:** 8
**Confidence:** 3

**Summary:**

Authors propose a training-free method for determining the optimal sliding window sizes for accelerating the inference speed of frozen Transformer language models (7B-70B parameters) trained using full attention. The authors demonstrate that using uniform window sizes across all model heads under a given sparsity budget leads to performance degradation and propose a method to optimally allocate window sizes to different heads. For each head h, they compute the approximate increase in model loss ΔL_hij resulting from prohibiting a query at position i from attending to a key at position j for all (i,j) pairs. This computation is efficient, depending only on ∂L/∂a_hij and a_hij's, where a_hij is the normalized attention score in the original model. The process requires just a single backward pass per sequence from a small calibration corpus. After determining ΔL_hij values, the authors compute optimal window size allocations per head by formulating a global multi-objective mixed integer programming (MIP) problem that satisfies the global sparsity budget while minimizing the sum of resulting ΔL_hij values. The MIPs can be solved efficiently using available solvers.

The authors' investigation of calibration corpus selection reveals that corpora involving long-range dependencies, such as summaries of long news articles, yield significantly better performance on evaluation tasks and improved retrieval accuracy.

At 50% sparsity, the authors report significant inference speedups with existing LLM frameworks such as FlashAttention2 and vLLM, achieved through various hardware optimizations including static KV-cache, larger batch sizes due to smaller KV cache, reduced attention computations due to sparsity, and specialized CUDA kernels.

The authors demonstrate that these speedups incur minimal performance loss and, compared to existing fast decoding frameworks H2O, infLLM, and StreamingLLM, their proposed method achieves significantly better long-context retrieval accuracy, long-context understanding scores on LV-Eval and LongBench, and perplexity on long-range text corpora.

**Strengths:**

1. The proposed method is elegant, mathematically principled, and efficient.

2. The method achieves significant decoding speedups with modest performance loss compared to strong baselines. Faster LLM inference is an important and timely topic of research.

3. The experiments are thorough and comprehensively support the authors' claims.

4. The authors have shared their codebase for reproducibility.

**Weaknesses:**

1. While the proposed method targets trained full-attention models, the need for heterogeneous window sizes might diminish with the emergence of uniform sliding window methods augmented with Mamba-like state space layers.

2. The methodology description, especially the MIP section, could be clearer.

**Questions:**

1. During profiling, what is the loss function? Is it the sum of entropy losses at all output positions or just the largest position? You mention that perplexity is evaluated only on the answer part - does this apply during profiling as well? Please clearly define the loss function.

2. Are the choices of (α_r), (β_r) pairs finite? If yes, what are they?

---

> ### Author Response · Authors · 2024-11-23
> **Rebuttal by Authors**
>
> ### [W1] SSM+Uniform attention may replace heterogeneous attention
>
> > While the proposed method targets trained full-attention models, the need for heterogeneous window sizes might diminish with the emergence of uniform sliding window methods augmented with Mamba-like state space layers.
>
> We thank the reviewer for raising such a new and interesting perspective and we wish to add a few personal options to it.
>
> We believe some insights that MoA provides may assist the design of future hybrid-attention designs. For example, MoA discovers that the first few layers require denser attention than its subsequent layers, as shown in Figure 11. Based on such observation, it may be sensible to use more transformer blocks in initial layers, and gradually add state-space blocks in subsequent layers. In addition, in the final layers, MoA finds that while most attention heads require low density, some high-density heads is still needed. Such result hints that it may be important to keep few transformer blocks at last layers, rather than replace them all with state-space blocks.
>
>
>
> ### [W2] Writing clarity for MIP section
>
> > The methodology description, especially the MIP section, could be clearer.
>
> We apologize for the confusion and revise Appendix D.3.2 with an example and figure illustration for better clarity.
>
>
>
> ### [Q1] Clarify: loss function at profiling
>
> > During profiling, what is the loss function? Is it the sum of entropy losses at all output positions or just the largest position? You mention that perplexity is evaluated only on the answer part - does this apply during profiling as well? Please clearly define the loss function.
>
> The loss function used during profiling is the sum of cross-entropy losses computed over the answer part generated by the original model.
> Using the answer part as supervision enhances model alignment and long-range dependency properties.
>
> We thank the reviewer for pointing out this potential confusion. To better connect the profiling stage discussed in Section 4.1 with the supervision details in Section 5, we have revised the paper to explicitly link the profiling loss definition to Section 5. Additionally, we have clarified the loss definition within Section 5 to improve overall clarity and coherence.
>
>
>
> ### [Q2] Clarify: search space for hyperpareter alpha and beta
>
> > Are the choices of ($\alpha$, $\beta$) pairs finite? If yes, what are they?
>
> Yes, the choices of ($\alpha$, $\beta$) pairs are finite. In our experiments, we use 6 values for $\alpha$ and 9 values for $\beta$, creating a search space of 54 pairs for each attention head. $\alpha$ is uniformly sampled from the range $[-2048, 8192]$, and $\beta$ is uniformly sampled from $[0,1]$. The resulting attention span lengths are clipped to the range between 0 and the current input length.
>
> We revise Section 3.2 and add hyperparameter details in Appendix A.1 for further clarity.

---

> ### Author Response · Authors · 2024-11-26
> **Looking forward to your reply**
>
> Dear Reviewer UNvV,
>
> We sincerely hope our revisions **clarify the presentation** of the method details and setups. If you have any remaining questions or require further clarification, please feel free to let us know at your earliest convenience. Once again, thank you for your insightful perspectives, valuable feedback, and contributions to improving our work.

---

> > ### Comment · Reviewer_UNvV · 2024-11-26
> > **Response to rebuttal 1**
> >
> > I thank the authors for including the above clarifications in the paper. Fig 15 now makes the MIP part clear. Overall, I found the contributions of this work to be substantial and believe that it should be accepted. I am maintaining my score.

---

> > > ### Author Response · Authors · 2024-11-27
> > > **Thank you for your affirmation**
> > >
> > > Thank you for your valuable suggestions and kind support! We are glad to have addressed your concerns and clarified more details about our work through your insightful feedback. Thank you again for your time and thoughtful review!

---

### Official Review · Reviewer_2JPU · 2024-11-04

**Soundness:** 3
**Presentation:** 3
**Contribution:** 3
**Rating:** 6
**Confidence:** 3

**Summary:**

The paper studies sparse attention configuration. Particuarlay, the authors identifies two hyper-parameters for sparse attention configuration, and adopts a gradient-based method to configure these two hyper-parameters in an adaptive manner. The author conduct experiments on various long context evaluation benchmarks with a 4k to 16k context length.

**Strengths:**

The studied problem is important and emerging. The proposed method is evaluated on LLms of various different sizes. Both wallclock time and memory consumption has been reported. The performance gain is consistent and significant.

**Weaknesses:**

My major concern is on the experiment design. As this study targets general sparse attention, it would be better to also conduct evaluation on popular LLM evaluations (e.g., AlpacaEval). Since the prevailing application for LLM is generation, it is necessary to inspect the potential performance degeneration on those tasks.

Also, I suggest the author to also evaluate the wallclock time and memory consumption for a longer context (e.g., 128k), since the proposed method should have the largest generation acceleration potential at those cases.

**Questions:**

Why the proposed method is named "Mixture of Attention"? I feel the name is a bit confusing and would suggest the author to change it to a more straightforward name.

---

> ### Author Response · Authors · 2024-11-23
> **Rebuttal by Authors**
>
> ### [W1] Add popular short-context benchmark: AlpacaEval
>
> > As this study targets general sparse attention, it would be better to also conduct evaluation on popular LLM evaluations (e.g., AlpacaEval). Since the prevailing application for LLM is generation, it is necessary to inspect the potential performance degeneration on those tasks.
>
> We thank the reviewer for the suggestion and have included the AlpacaEval 2.0 benchmark in Appendix B.1.5. As shown in Table 10, MoA achieves the highest length-controlled win rate, outperforming other sparse attention baselines and even the original dense model.
>
> |Attention|Length-controlled Win Rate|Standard Error|
> |-|-|-|
> |Original|8.84|0.53|
> |H2O|9.66|0.55|
> |InfLLM|5.76|0.42|
> |StreamingLLM|7.96|0.49|
> |MoA|**9.83**|0.57|
>
> We would also like to note that our work mainly targets the long-context inference scenario, where the attention computation and kv cache storage is heavy. Therefore, we mainly adopt long-context benchmarks, instead of benchmarks with shorter sequences (e.g., approximately 50 tokens of input and 450 tokens of output in AlpacaEval).
>
> Nevertheless, we agree that the addition of AlpacaEval can strengthen the evidence for MoA’s generalizability and robust performance across a wide range of input and output lengths.
>
>
> ### [W2] Efficiency results at 128K length
>
> > Also, I suggest the author to also evaluate the wallclock time and memory consumption for a longer context (e.g., 128k), since the proposed method should have the largest generation acceleration potential at those cases.
>
> We thank the reviewer for the valuable suggestion and have added the 128K efficiency test results in Appendix B.2.2.
> Thanks to the reduced KV-Cache, MoA efficiently processes 128k input using only one A100 GPU, whereas FlashAttention2 and vLLM baselines require at least two GPUs to handle a single request.
> As shown in the Table, MoA achieves a $4.7$-$4.9\times$ decode speedup compared to FlashAttention2, while using half the number of GPUs. Additionally, it demonstrates a $1.9$-$2.1\times$ reduction in GPU memory usage.
> Compared to vLLM, which utilizes tensor parallelism, MoA delivers $1.3$-$1.4\times$ higher throughput per GPU, alongside significant memory savings.
>
>
> |Model Size|Framework|Attention|Min. #GPU|Total Throughput|Total Memory(GB)|Throughput per GPU|
> |-|-|-|-|-|-|-|
> |**7B**|vLLM|PagedAttention|2|30.2|142.0|15.1|
> ||FlexGen|H2O|>8|-|OOM|-|
> ||HuggingFace|InfLLM|1|6.1|47.7|6.1|
> ||HuggingFace|StreamingLLM|1|19.8|43.9|19.8|
> ||HuggingFace|FlashAttention2|2|4.3|85.6|2.2|
> ||HuggingFace|MoA|1|20.3|44.0|20.3|
> |**13B**|vLLM|PagedAttention|2|21.5|142.0|10.8|
> ||FlexGen|H2O|>8|-|OOM|-|
> ||HuggingFace|InfLLM|1|4.3|78.6|4.3|
> ||HuggingFace|StreamingLLM|1|14.0|64.6|14.0|
> ||HuggingFace|FlashAttention2|2|3.0|130.6|1.5|
> ||HuggingFace|MoA|1|14.7|63.4|14.7|
>
>
> ### [Q1] The reason for the name: MoA
>
> > Why the proposed method is named "Mixture of Attention"? I feel the name is a bit confusing and would suggest the author to change it to a more straightforward name.
>
> We thank the reviewer for raising this discussion. The name "Mixture of Sparse Attention" (MoA) was chosen to highlight the importance of heterogeneity in sparse pattern designs. The core idea of our method is to mix different attention spans and elastic rules across attention heads, aligning with their distinct roles within the model. Inspired by "Mixture of Experts" (MoE), the name also reflects the diversity of attention heads' functions in LLMs and aims to encourage further exploration in this area.
>
> We apologize for any confusion the name may have caused. If this explanation does not fully address your concern, we are open to discussing alternative names that better capture the method’s contributions.

---

> ### Author Response · Authors · 2024-11-26
> **Looking forward to your reply**
>
> Dear Reviewer 2JPU,
>
> We sincerely hope we address your concerns by adding the **AlpacaEval benchmark** and **efficiency tests at 128k**. We look forward to your feedback on these updates. If there are any remaining questions, please let us know at your earliest convenience. We are also happy to engage in further discussions regarding the name of the paper. Once again, thank you for your valuable time and contributions to refining our work.

---

> > ### Comment · Reviewer_2JPU · 2024-12-01
> > **Response to rebuttal 1**
> >
> > Thanks the author for the new results. These results addressed my concerns, and I am maintaining my score. Additionally, on a side node, another suggestion is to also include the win-rate w.r.t. the dense model in the W1 results.

---

> > > ### Author Response · Authors · 2024-12-02
> > > **Thank you for your affirmation**
> > >
> > > Thank you for your positive feedback and affirmation of our rebuttal. Based on your suggestion, we have conducted an additional experiment to include the win-rate with respect to the original dense model. The following table will be included in the final manuscript.
> > >
> > > Thank you once again for your time and thoughtful review! We hope the additional results enhance our paper and reflect positively in your score.
> > >
> > > > **Table: AlpacaEval 2.0 Length-Controlled Win Rate and Standard Error Over the Original Dense Model.**
> > >
> > > |Attention|Length-controlled Win Rate|Standard Error|
> > > |-|-|-|
> > > |H2O|47.90|0.39|
> > > |InfLLM|44.76|0.36|
> > > |StreamingLLM|46.47|0.47|
> > > |MoA|**48.39**|0.47|

---

### Official Review · Reviewer_9X5F · 2024-11-12

**Soundness:** 3
**Presentation:** 3
**Contribution:** 2
**Rating:** 3
**Confidence:** 5

**Summary:**

The paper proposes the Mixture of Attention (MoA), a novel approach for compressing Large Language Models (LLMs) by leveraging sparse attention. Traditional sparse attention methods apply uniform sparse attention masks across all heads and input lengths, limiting the model’s effectiveness due to the lack of diversity in attention spans. MoA addresses this limitation by introducing heterogeneous sparse attention configurations across different attention heads and layers, allowing the model to optimize its focus on both local and global contexts. This approach extends the effective context length, enhances retrieval accuracy, and reduces GPU memory usage. The authors demonstrate that MoA can increase the context length by 3.9 times and improve retrieval accuracy by 1.5-7.1 times, while achieving significant gains in throughput and memory efficiency compared to other sparse attention baselines.

**Strengths:**

MoA's introduction of heterogeneous elastic rules across different heads and layers allows for more targeted attention configurations, aligning with the diversity of information retrieval needs across various input lengths. This significantly improves retrieval accuracy and extends the effective context length, providing clear benefits over uniform sparse attention models. MoA features an automated pipeline to determine optimal sparse configurations tailored for each model and dataset, based on profiling each attention head's impact on prediction loss. This approach enhances the model's adaptability to specific datasets with long-range dependencies, demonstrating impressive scalability and performance benefits without the need for retraining. The method achieves substantial GPU memory reduction and throughput improvements, showing a notable advantage in environments with limited computational resources. With up to 8.2 times improvement in throughput and minimal retrieval accuracy loss, MoA represents a valuable tool for deploying LLMs on a larger scale with reduced computational burden. The paper presents a comprehensive evaluation across several models and benchmarks, confirming MoA’s efficiency and effectiveness in diverse settings. Its performance on Vicuna and Llama3 models at different input lengths reinforces the robustness and scalability of the proposed method.

**Weaknesses:**

1. Sparse attention has been extensively explored in prior research, notably in [1] and [2], which diminishes the novelty of this study. Additionally, H2O [3] has already thoroughly examined the feedback from using a sliding window approach.
2. There is a lack of comparative baselines, specifically H2O [3], SnapKV [4], and PyramidKV [5]. Including these would provide a clearer assessment of the proposed method.
3. An evaluation of Needle is necessary to demonstrate the motivation behind preserving long-range dependencies effectively.
4. The proposed method is really similar to a previous method RazorAttention [6]. I recommend adding this to citations.




[1] https://arxiv.org/abs/2402.17762
[2] https://arxiv.org/pdf/2309.17453
[3] https://arxiv.org/abs/2306.14048
[4] https://arxiv.org/abs/2404.14469
[5] https://arxiv.org/abs/2406.02069
[6] https://arxiv.org/abs/2407.15891

**Questions:**

As weakness

---

> ### Author Response · Authors · 2024-11-23
> **Rebuttal by Authors**
>
> ### [W1.1] Difference between prior sparse attention works [1-3]
>
> > Sparse attention has been extensively explored in prior research, notably in [1] and [2], which diminishes the novelty of this study.
>
> We acknowledge the extensive prior works on **uniform** sparse attention. They primarily address the **coherence problem** in long contexts. Instead, our approach explores the **heterogeneous** elastic rules for attention patterns. We further address the distinct challenge of extending the **effective context length** of LLMs. Specifically:
>
> * Prior work [1] analyzes massive activations in LLMs, revealing the sparse nature of attention.
>
> * StreamingLLM [2] proposes uniform sparse pattern and attention sink, improving response coherence, especially for streaming scenarios and out-of-distribution lengths.
>
> * H2O [3] proposes uniform dynamic sparse pattern, reducing KV-Cache during the decoding stage.
>
> * Our work proposes heterogeneous sparse patterns and their elastic rules, uniquely expanding the effective context length of LLMs.
>
>
> ### [W1.2] Sliding window approach has been explored by H2O [3]
>
> > Additionally, H2O [3] has already thoroughly examined the feedback from using a sliding window approach.
>
> We wish to clarify that the sliding window pattern is not our contribution. Instead, we construct and optimize heterogeneous elastic rules for this pattern across heads and lengths. MoA shows significant improvements in both accuracy and efficiency (see Figures 1, 4, 5 and Tables 4, 5 in the main paper). Our results demonstrate clear advantages over [2] and [3], which serve as our primary baselines.
>
>
> ### [W2] Add comparative baselines H2O [3], SnapKV [4], and PyramidKV [5]
>
> > There is a lack of comparative baselines, specifically H2O [3], SnapKV [4], and PyramidKV [5].
>
> We thank the reviewer and add all the queried baselines. As shown in the updated Figure 4, MoA consistently achieves a superior throughput-accuracy trade-off across varied densities, outperforming all six baselines.
> Additionally, we include the long context retrieval performance of SnapKV [4] and PyramidKV [5] in the following Table. MoA shows 4% to 21% higher retrieval accuracy compared to [4,5] for 64K to 256K lengths. H2O [3] encounters OOM errors at these lengths.
>
> |Method|32k|64k|128k|256k|
> |-|-|-|-|-|
> |Original|0.98|0.93|0.76|0.37|
> |-|-|-|-|-|
> |SnapKV|**1.00**|0.88|0.71|0.33|
> |PyramidKV|**1.00**|0.85|0.62|0.37|
> |MoA|**1.00**|**0.92**|**0.83**|**0.46**|
>
> We also wish to note that H2O [3] is one of the primary baselines in our work, with comprehensive comparisons provided in Figures 4 and 5, as well as Tables 4 and 5 of the main paper.
>
>
> ### [W3] Add Needle-In-A-Haystack (NIAH) evaluation
>
> > An evaluation of Needle is necessary to demonstrate the motivation behind preserving long-range dependencies effectively.
>
> We add the Needle-In-A-Haystack (NIAH) retrieval evaluation [7] in Appendix B.1.2, Figure 8. As shown in the Figure, MoA shows perfect retrieval accuracy across 8k to 256k lengths.
>
> Note that our primary benchmark LongEval[8] also validates the retrieval ability of models, which are shown in Figure 1,4,5 and Table 1,3,4 of the main paper. The result of NIAH evaluation aligns with the results observed in the LongEval benchmark.
>
> ### [W4] Reference concurrent work: RazorAttention [6]
>
> > The proposed method is really similar to a previous method RazorAttention [6]. I recommend adding this to citations.
>
> We thank the reviewer and discuss the concurrent work RazorAttention[6] in Appendix E. Both MoA and RazorAttention use different strategies for different attention heads. They mainly distinguish from each other in the following aspects:
>
> 1. **Strategies for attention heads**: RazorAttention categorizes attention heads into two types: retrieval and non-retrieval. It adopts bipolar strategies, applying either full attention or fixed-sized local attention. In contrast, MoA recognizes the diverse attention spans of different heads and employs a broader range of strategies, covering the entire spectrum from very limited local attention to full attention.
>
> 2. **Adaptation to different input lengths**: RazorAttention uses a fixed density for non-retrieval heads across all input lengths. MoA, on the other hand, applies heterogeneous elastic rules for each head, dynamically adjusting densities based on input length while maintaining overall density constraints.
>
> 3. **Determination of strategies**: RazorAttention relies on a heuristic approach to assign strategies. It uses attention scores between the current token and specific tokens (e.g., echo and induction tokens) to identify retrieval heads and assign full attention. MoA employs a loss-based method. It uses a first-order Taylor expansion to estimate the impact of a strategy on end-to-end prediction loss and optimizes strategies to minimize this loss under density constraints.
>
> By discussing the concurrent work RazorAttention [6], we provide a more comprehensive context for the paper.

---

> > ### Author Response · Authors · 2024-11-23
> > **References**
> >
> > ### References
> >
> > #### Mentioned by the reviewer
> >
> > [1, Massive Activations in Large Language Models] https://arxiv.org/abs/2402.17762
> >
> > [2, StreamingLLM] https://arxiv.org/pdf/2309.17453
> >
> > [3, H2O] https://arxiv.org/abs/2306.14048
> >
> > [4, SnapKV] https://arxiv.org/abs/2404.14469
> >
> > [5, PyramidKV] https://arxiv.org/abs/2406.02069
> >
> > [6, RazorAttention] https://arxiv.org/abs/2407.15891
> >
> > #### Additional
> >
> > [7] Greg Kamradt. Llmtest_needleinahaystack: Doing simple retrieval from llm models at various context lengths to measure accuracy. https://github.com/gkamradt/LLMTest_NeedleInAHaystack, 2024. Accessed: 2024-11-18.
> >
> > [8] Dacheng Li, Rulin Shao, Anze Xie, Ying Sheng, Lianmin Zheng, Joseph E. Gonzalez, Ion Stoica, Xuezhe Ma, and Hao Zhang. How long can open-source llms truly promise on context length?, June 2023. URL https://lmsys.org/blog/2023-06-29-longchat

---

> ### Author Response · Authors · 2024-11-26
> **Looking forward to your reply**
>
> Dear Reviewer 9X5F,
>
> We sincerely hope our revisions address your major concerns regarding **prior works and baselines**. We look forward to your feedback on these updates. If there are any remaining questions, please let us know at your earliest convenience. We would be happy to engage in further discussions. Once again, thank you for your valuable time and contributions to refining our work.

---

> > ### Author Response · Authors · 2024-11-30
> > **Polite reminder to response**
> >
> > Dear Reviewer 9X5F,
> >
> > As the rebuttal period is nearing its conclusion, we would like to kindly remind you that we have not yet received your feedback on our revised manuscript, which includes all the additional baselines and experiments you requested. Should you have any further questions or require additional clarification, please do not hesitate to contact us. Thank you once again for your time and suggestions.

---

> ### Comment · Reviewer_9X5F · 2024-12-03
>
> Thanks for the authors' efforts. I have several questions and concerns to the authors:
>
> 1. What is the main difference from MoA's heterogeneous sparse patterns to other paper that also discusses about the attention sparsity? For example, [1]'s method is head-level adaptive compression, [2]'s method also identify the unique patterns of diverse heads. I think [1] and [2] both apply the uniform sparse attention across heads, and then build methods based on these patterns. I am not pretty sure about the unique novelty of heterogeneous elastic rules for attention patterns from MoA.
>
> 2. It may be also helpful to discuss MoA with [3], [3] also identify diverse unique patterns in long-context attention, which is a close setting compared with MoA.
>
> Since the rebuttal period is close to the end, authors could just briefly answer these questions without empirical results. Thanks!
>
> [1] SnapKV: LLM Knows What You are Looking for Before Generation
> [2] MODEL TELLS YOU WHAT TO DISCARD: ADAPTIVE KV CACHE COMPRESSION FOR LLMS
> [3] MInference 1.0: Accelerating Pre-filling for Long-Context LLMs via Dynamic Sparse Attention

---

> > ### Author Response · Authors · 2024-12-03
> > **Discussion on related works**
> >
> > We thank the reviewer's question. Below, we highlight MoA’s main differences compared to the referenced works in the following table (major differences are highlighted in **bold**).
> >
> > Due to time constraints during the rebuttal period, the decode throughput values are derived using latency and speedup numbers reported in the respective papers. The setup uses 7B LLMs with 4k input length, batch size 8, 50% density, running on a single A100 GPU. Win rates are as reported in the respective papers.
> >
> >
> > | Work                             | Stage       | Category         | Heterogeneous vs. Uniform | Search Space                                                 | Generalize to Longer Input | Search Method                                    | Decode Throughput (tokens/s)        | Win Rate Over Dense |
> > |----------------------------------|-------------|------------------|---------------------------|-------------------------------------------------------------|-----------------------------|-------------------------------------------------|--------------------------------------|---------------------|
> > | [1] SnapKV                       | Decode      | Dynamic          | **Uniform**               | -                                                           | $\times$                   | -                                               | 83 (calculated with decode latency in paper)       | NA                  |
> > | [2] FastGen                      | Decode      | Dynamic + Static | Heterogeneous             | **Handcrafted heuristics** of combinations of **token categories** | $\times$                   | **Independent** greedy search for each head     | 118 (estimated with maximum speedup over DS)  | 30.8\%              |
> > | [3] MInference (concurrent work) | **Prefill** | Dynamic + Static | Heterogeneous             | Lambda-shape, vertical-slash, and block-sparse patterns     | $\times$                   | **Independent** greedy search for each head     | 76 (same as dense)                   | NA                  |
> > | **MoA**                          | Decode      | **Static**       | Heterogeneous             | Attention span's **elastic rules w.r.t. input length**         | **$\checkmark$**            | **Joint** mixed-integer programming across heads | 260 (tested)                         | 48.4\%              |
> >
> >
> > As shown in the table, MoA differs from previous works in the following key aspects:
> >
> > **1. Search space:** MoA uniquely incoporates the heterogeneous elastic rule of attention patterns **with respect to input length**. It opens up new search spaces and enabling **generalization to longer inputs** that has not been demonstrated by previous works. By compressing on a calibration dataset within 12k, MoA directly generalizes to input lengths **up to 256k** (Figure 5).
> > Additionally, MoA employs **static attention patterns**, are predetermined and do not require per-sequence computation, incurring **no additional computational overhead** during prefilling. As a result, MoA achieves significantly higher decode throughput compared to the mentioned methods.
> >
> > **2. Search method:** MoA uniquely considers accuracy-efficiency trade-offs **across all heads** through **gradient-based loss estimation**. It enables **joint optimizations** with MIP-based automatic compression pipeline. In contrast, previous works perform **independent optimizations** for **individual heads** using greedy search, which minimizes recall for each attention matrix but does not optimize end-to-end prediction. Thus, MoA achieves higher retrieval accuracy (Table 9) and a superior win rate.
> >
> > We hope our responses address the reviewer's questions about additional related works. We are happy to answer any further questions.

---

### Author Response · Authors · 2024-11-23
**Overall Response**

We sincerely thank all reviewers for their valuable time and insightful feedback. We are encouraged that all reviewers recognize the “consistent”, “significant,” and “impressive” performance improvements achieved by our heterogeneous sparse attention design. It is motivating to see the reviewers appreciate the “important” and “timely” nature of the topic (Reviewers 2JPU, UNvV, pLr2), describe our method as “elegant and mathematically principled” (Reviewer UNvV), and commend our “thorough” and “comprehensive” experiments (Reviewers 9x5F, UNvV), which demonstrate “substantial” efficiency benefits (Reviewers 9X5F, UNvV) “compared to strong baselines” (Reviewer UNvV).

In response to the reviewers’ comments, we have made the following improvements in the revised paper:

1. **Two additional baselines.** We have updated Figure 4 to include two additional baselines, SpanKV and PyramidKV, illustrating the throughput-accuracy trade-off across varied densities. MoA consistently advances the Pareto front beyond all six baselines, demonstrating superior performance.

2. **Two additional benchmarks.** We add the short-context long-output generation benchmark AlpacaEval 2.0 in Appendix B.1.5 and another Needle-In-A-Haystack long-context retrieval benchmark in Appendix B.1.2. Alongside LongEval, LongBench, LV-Eval benchmarks, and various perplexity tests, these additions reinforce MoA’s superior performance across various tasks.

3. **Efficiency results at 128K.** We add efficiency experiments at a longer context length of 128K in Appendix B.2.2, further showcasing MoA’s scalability and efficiency benefits in handling extremely long sequences.

4. **Improved writing.** We improve the writing to further clarify details such as the profiling loss function definition, search space hyperparameters, and the detailed mixed-integer-programming formulation. We also add more discussions and references to concurrent works for a more comprehensive context. Major revisions are marked with blue color.

---

### Meta-Review · Area_Chair_4BXt · 2024-12-18

**Metareview:**

The paper proposes the Mixture of Attention (MoA), an approach for compressing Large Language Models (LLMs) by leveraging sparse attention.

Reviwers provide very mixed score with 8, 6, 5, and 3.
AC carefully read the paper, the reviewers' comments and authors' response.

While two reviewers gave positive scores, the main concerns of two reviewers seem the limitation in practical uses in real-world.

Considering the ICLR competitiveness, AC should recommends rejecting this paper in this time.

**Additional Comments On Reviewer Discussion:**

This paper's initial scores were 8, 6, 5 and 3. Although there were some discussion between the authors and reviewers, all reviewers did not change their score. Two reviewers with negative score argued their main concerns still remained.

---

### Decision · Program_Chairs · 2025-01-22

Reject